# ASAR lncRNAs control DNA replication timing through interactions with multiple hnRNP/RNA binding proteins

**Mathew Thayer[1]\*[†], Michael B Heskett[2,3][†], Leslie G Smith[1], Paul T Spellman[2,4][‡], Phillip A Yates[1]**

[1]Department of Chemical Physiology and Biochemistry,Oregon Health & Science University, Portland, United States; [2]Department of Molecular and Medical Genetics, Oregon Health & Science University, Portland, United States; [3]Stanford Cancer Institute, Stanford, United States; [4]Cancer Early Detection Advanced Research Center, Knight Cancer Institute, Oregon Health & Science University, Portland, United States

**\*For correspondence:**
thayerm@ohsu.edu

[†]These authors contributed equally to this work

**Present address:** [‡]Departments of Medicine and Human Genetics, University of California, Los Angeles, Los Angeles, United States

**Competing interest:** The authors declare that no competing interests exist.

**Abstract** ASARs are a family of very-long noncoding RNAs that control replication timing on individual human autosomes, and are essential for chromosome stability. The eight known ASAR lncRNAs remain closely associated with their parent chromosomes. Analysis of RNA-protein interaction data (from ENCODE) revealed numerous RBPs with significant interactions with multiple ASAR lncRNAs, with several hnRNPs as abundant interactors. An ~7 kb domain within the *ASAR6-141* lncRNA shows a striking density of RBP interaction sites. Genetic deletion and ectopic integration assays indicate that this ~7 kb RNA binding protein domain contains functional sequences for controlling replication timing of entire chromosomes in cis. shRNA-mediated depletion of 10 different RNA binding proteins, including HNRNPA1, HNRNPC, HNRNPL, HNRNPM, HNRNPU, or HNRNPUL1, results in dissociation of ASAR lncRNAs from their chromosome territories, and disrupts the synchronous replication that occurs on all autosome pairs, recapitulating the effect of individual ASAR knockouts on a genome-wide scale. Our results further demonstrate the role that ASARs play during the temporal order of genome-wide replication, and we propose that ASARs function as essential RNA scaffolds for the assembly of hnRNP complexes that help maintain the structural integrity of each mammalian chromosome.

## eLife assessment

This **important** study expands generally upon our understanding of the role of hnRNP proteins in lncRNA function through analysis of ASAR genes that are present on all chromosomes and of profound significance. The findings provide **convincing** evidence linking ASARs with the phenomenon of RNA retention on chromosomes, including X inactivation, thereby providing an expanded context for studies in these areas. This manuscript will be of interest to researchers studying gene regulation and the interactions and functional roles of hnRNP and lncRNAs.

## Introduction

The vast majority of mammalian DNA replicates in homologous regions of chromosome pairs in a highly synchronous manner (*Mukhopadhyay et al., 2014*; *Dileep and Gilbert, 2018*; *Rivera-Mulia et al., 2018*). However, genetic disruption of non-protein coding ASAR ('*AS*ynchronous replication and *Autosomal RNA*') loci causes a delay in replication timing on individual human chromosomes in

cis, resulting in highly asynchronous replication between pairs of autosomes (*Donley et al., 2015*; *Heskett et al., 2020*; *Heskett et al., 2022*; *Stoffregen et al., 2011*). There are eight genetically validated ASARs that are located on human chromosomes 1, 6, 8, 9, and 15 (*Donley et al., 2015*; *Heskett et al., 2020*; *Heskett et al., 2022*; *Stoffregen et al., 2011*). All of the known ASAR lncRNAs share several distinctive characteristics, including: (1) contiguously transcribed regions of >180 kb, (2) epigenetically regulated allelic expression imbalance (AEI; including mono-allelic expression); (3) variable epigenetic replication timing (VERT; including asynchronous replication timing); (4) high density of LINE1 (L1) sequences (>30%); and (5) retention of the RNA within their parent chromosome territory. Recently, we used these notable characteristics to identify 68 ASAR candidates on every human autosome (*Heskett et al., 2020*; *Heskett et al., 2022*).

In addition to genetic disruption assays, ectopic integration of ASAR transgenes into mouse chromosomes has been used as a second functional assay to help define the critical sequences within ASAR lncRNAs that control chromosome-wide replication timing (*Donley et al., 2013*; *Platt et al., 2018*). For example, *ASAR6* lncRNA expressed from transgenes, ranging in size from an ~180 kb BAC transgene to PCR products (as small as ~3 kb) containing the critical sequences, remain associated with the chromosome territories where they are transcribed (*Donley et al., 2013*; *Platt et al., 2018*). These same transgenes cause delayed replication timing (DRT) and delayed mitotic condensation (DMC) of entire chromosomes in cis (*Donley et al., 2013*; *Platt et al., 2018*). Furthermore, *ASAR6* lncRNA was found to control chromosome-wide replication timing using oligonucleotide-mediated RNA degradation of *ASAR6* lncRNA that was expressed from a BAC transgene integrated into mouse chromosome 3 (*Platt et al., 2018*). These findings suggest that ASARs represent ubiquitous, essential, *cis*-acting elements that control mammalian chromosome replication timing via expression of chromosome associated RNAs. In this report, we identify RNA binding proteins (RBPs) that are critical for ASAR lncRNA localization and function.

Chromosome-associated long non-coding RNAs are known to be important for numerous aspects of chromosome dynamics in diverse eukaryotes from yeast to mammals (*Ding et al., 2019*; *Faber and Vo, 2022*; *Nozawa and Gilbert, 2019*; *Creamer et al., 2021*; *Menon and Meller, 2015*). The mammalian nucleus is rich in heterogenous nuclear RNA (hnRNA), which represents >95% of the total RNA Polymerase II output (*Nozawa and Gilbert, 2019*; *St Laurent et al., 2012*). The vast majority of hnRNA is poorly defined, has been referred to as the 'Dark Matter' of the genome (*St Laurent et al., 2012*), and includes spliced and unspliced intronic sequences, circular RNAs (*Kristensen et al., 2019*), very long intergenic non-coding RNAs (vlincRNAs; *St Laurent et al., 2016*), as well as poorly defined RNAs that include chromatin associated RNAs (caRNAs; *Nozawa et al., 2017*) and *re*peat-rich RNAs (repRNAs; *Frank and Rippe, 2020*) that can be detected by C0T-1 DNA (which is comprised primarily of LINE and SINE repetitive sequences; *Creamer et al., 2021*; *Hall et al., 2014*; *Kolpa et al., 2022*). Some of these nuclear RNAs comprise abundant species, yet lack clearly defined functions, while others are thought to be the by-products of other nuclear processes and have been collectively called 'RNA debris' (*Nozawa and Gilbert, 2019*). The relationship between ASAR lncRNAs and these other chromosome-associated RNAs remains poorly defined.

Previous studies have suggested that long nascent transcripts that are detected by C0T-1 DNA play a dynamic structural role that promotes the open architecture of active chromosome territories (*Creamer et al., 2021*; *Hall et al., 2014*). The RNA species detected by C0T-1 DNA are predominantly L1 sequences, and these L1 RNAs remain associated with the chromosome territories where they are transcribed (*Hall et al., 2014*). A link between C0T-1 RNA and ASAR lncRNAs is suggested by the observation that ASAR lncRNAs contain a high L1 content and remain associated with their parent chromosome territories (*Donley et al., 2015*; *Heskett et al., 2020*; *Heskett et al., 2022*; *Stoffregen et al., 2011*; *Platt et al., 2018*). In addition, deletion and ectopic integration assays demonstrated that the critical sequences for controlling chromosome-wide replication timing within *ASAR6* and *ASAR15* map to the antisense strand of L1 sequences located within the *ASAR6* and *ASAR15* lncRNAs (*Platt et al., 2018*). In this report we found that human C0T-1 DNA can detect the chromosome territory localization of *ASAR6* lncRNA that is expressed from a BAC transgene integrated into a mouse chromosome, indicating that at least some of the RNA species detected by C0T-1 DNA represents ASAR lncRNA.

Heterogeneous nuclear ribonucleoproteins (hnRNPs) represent a large family of abundant RBPs that contribute to multiple aspects of nucleic acid metabolism. Many of the proteins that are in hnRNP

complexes share general features, but differ in domain composition and functional properties (reviewed in *Geuens et al., 2016*). The functions of hnRNPs vary according to their subcellular localization. Most of the hnRNPs possess conventional nuclear localization signals and are predominantly present in the nucleus during steady state, where they function in various RNA metabolic processes including transcription, splicing, nuclear export, and 3' end processing (*Geuens et al., 2016*; *Domanski et al., 2022*), and more recently have been implicated in DNA replication (*Connolly et al., 2022*). A direct connection between chromosome territory associated C0T-1 RNA and hnRNPs was established by the observation that siRNA mediated depletion, or forced expression of dominant interfering mutants, of *HNRNPU* results in dissociation of C0T-1 RNA from chromosome territories (*Creamer et al., 2021*; *Hall et al., 2014*; *Kolpa et al., 2022*). In addition, recent studies have proposed that abundant nuclear proteins such as HNRNPU nonspecifically interact with 'RNA debris' that creates a dynamic nuclear mesh that regulates interphase chromatin structure (*Nozawa and Gilbert, 2019*; *Nozawa et al., 2017*). Here, we used publicly available enhanced Cross Link Immuno-Precipitation (eCLIP) data from ENCODE (*Moore et al., 2020*) to identify RBPs, including HNRNPA1, HNRNPC, HNRNPL, HNRNPM, HNRNPU, and HNRNPUL1, that interact with the known ASAR lncRNAs. We also found that an ~7 kb region within the ~185 kb *ASAR6-141* lncRNA has a surprisingly high density of interacting RBPs, and that genetic deletion of this ~7 kb 'RBP domain' from the endogenous *ASAR6-141* locus results in a delayed replication timing phenotype that is comparable to deletion of the entire ~185 kb *ASAR6-141* transcribed region located on human chromosome 6. In addition, ectopic integration of trans-genes, expressing the ~7 kb RBP domain, into autosomes or into the inactive X chromosome results in retention of the RNA within the chromosome territory of the integrated chromosomes, and results in DRT/DMC, and chromosome structure instability in cis. Taken together, these results indicate that the ~7 kb RBP domain represents a critical region within *ASAR6-141* that controls replication timing of human chromosome 6.

Because ASARs were identified as *cis*-acting elements that control chromosome-wide replication timing, we tested if ASAR associated RBPs also control replication timing. We found that shRNA mediated depletion of 9 different ASAR-associated RBPs (HNRNPA1, HNRNPC, HNRNPL, HNRNPM, HNRNPU, HNRNPUL1, HLTF, KHSRP, or UCHL5) dramatically altered the normally synchronous repli-cation timing program of all autosome pairs, recapitulating the effect of individual ASAR knockouts on a genome-wide scale. These results demonstrate the role that ASARs play during the temporal order of genome-wide replication, and suggests that ASAR lncRNAs serve as essential RNA scaffolds for the assembly of multiple hnRNP/RBP complexes that function to maintain the structural integrity of each mammalian chromosome.

## Results

All of the eight genetically validated ASARs are subject to AEI and express lncRNAs that remain associated with the chromosome territories where they are transcribed (*Donley et al., 2015*; *Heskett et al., 2020*; *Heskett et al., 2022*; *Stoffregen et al., 2011*). Examples of mono-allelic expression and chromosome territory localization of *ASAR6-141* lncRNA are shown in *Figure 1A and B*, where *ASAR6-141* lncRNA is localized within one of the chromosome 6 territories in two different cell types, male HTD114 cells and female GM12878 cells, respectively. We note that the size of the RNA hybrid-ization signals detected by all ASAR probes are variable, ranging in size from large clouds that occupy entire chromosome territories, with similarity in appearance to XIST RNA 'clouds' (*Figure 1B*), to relatively small sites of hybridization that remain tightly associated with the expressed alleles (*Donley et al., 2015*; *Heskett et al., 2020*; *Heskett et al., 2022*; *Stoffregen et al., 2011*) .

We first sought to identify RBPs that interact with the known ASAR lncRNAs. Using publicly avail-able eCLIP data for 150 RBPs in K562 and HepG2 cells (ENCODE; *Moore et al., 2020*; *Minks and Brown, 2009*), we identified RBP interactions within 4 ASARs (*ASAR1-187*, *ASAR6-141*, *ASAR8-2.7*, and *ASAR9-23*). Utilizing the eCLIP peaks previously identified by ENCODE (*Moore et al., 2020*) we found significant RBP binding peaks for 99 different RBPs that map within the four ASAR lncRNAs (*Figure 1—source data 1*). We also utilized a 'region-based' method (*Van Nostrand et al., 2016*) to identify significant enrichments of eCLiP reads against matched controls in 10 kb windows across the four ASAR loci (*Figure 1—source data 2*). An example of this analysis is shown in *Figure 1C*, where *ASAR6-141* displays significant RBP interactions for 35 different RBPs across the transcribed region. Given the contiguous expression (RNAseq from total RNA) across the ~185 kb *ASAR6-141* locus in

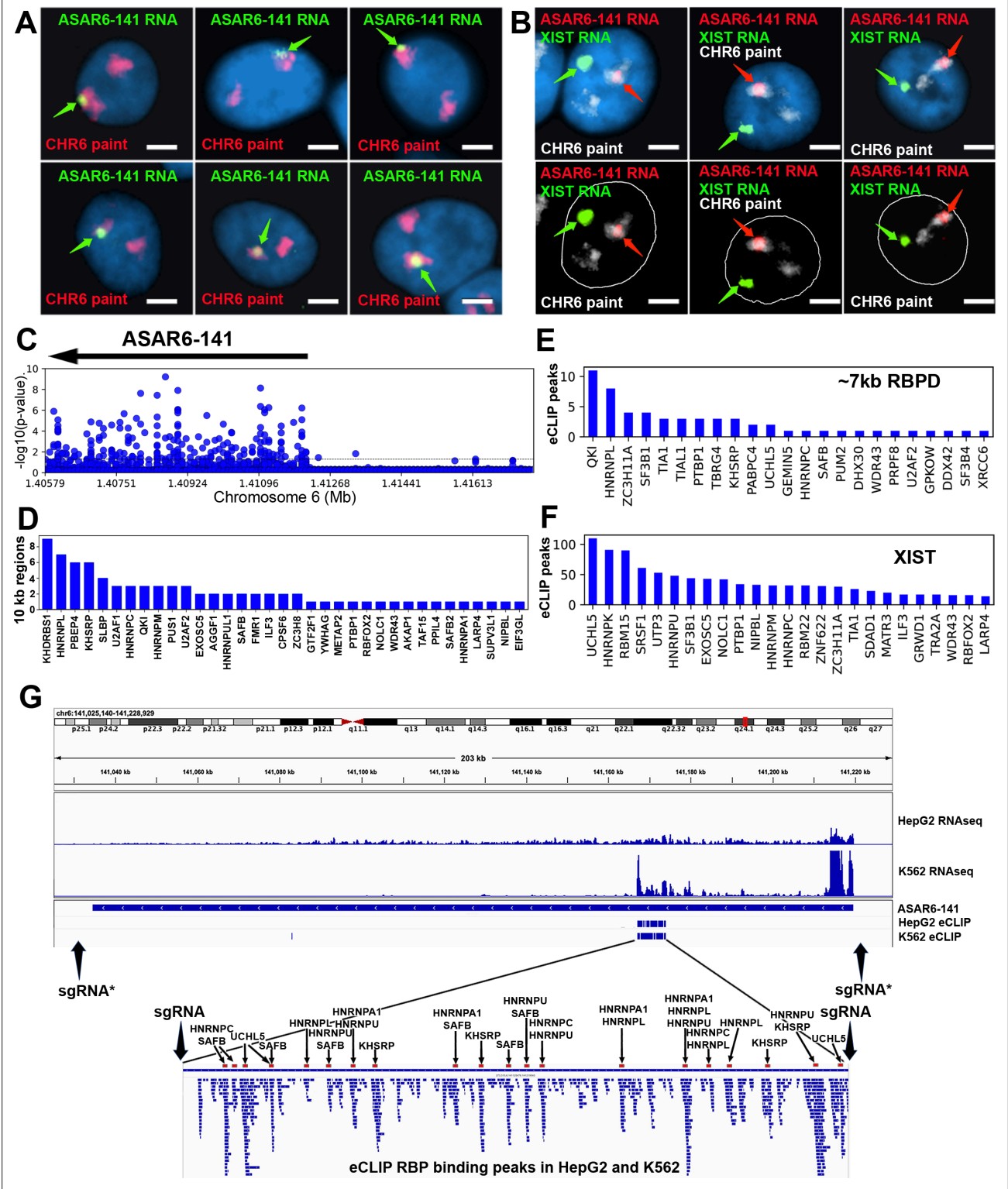

**Figure 1.** RNA binding proteins interact with an ~7 kb domain within ASAR6-141 RNA. (**A**) RNA-DNA FISH images of *ASAR6-141* expression in six individual HTD114 cells. ASAR6-141 RNA (green; green arrows) and chromosome 6 DNA (chromosome paint, red), and DNA was stained with DAPI. (**B**) RNA-DNA FISH of *ASAR6-141* RNA (red, red arrows), CHR6 DNA (chromosome paint, white), and XIST RNA (green, green arrows) visualized within individual GM12878 cells, top and bottom panels represent the same three cells with the nuclear outline drawn in white. (**C**) RNA-protein interaction data (eCLIP from ENCODE) for 120 RBPs expressed in K562 cells for the genomic region that contains *ASAR6-141*, and ~200 kb of upstream non-transcribed DNA (chromosome 6 140.8 mb-141.6 mb). Each point represents the FDR-corrected p-value of the z-score of log2-ratio between eCLiP vs

*Figure 1 continued on next page*

*Figure 1 continued*

control for each RBP within a 10 kb sliding window. (**D**) Number of 10 kb regions with significant enrichments of eCLiP reads vs control within *ASAR6-141*. (**E**) Histogram of eCLiP peaks count per RBP within *ASAR6-141* 7 kb RBPD. (**F**) Histogram of eCLiP peaks count per RBP within lncRNA XIST. (**G**) Genome browser view of *ASAR6-141* with RNA-seq expression and eCLiP peaks shown in K562 and HepG2. The zoomed in view shows the ~7 kb RBPD with the location of the peaks of eCLIP reads that map within the region (see *Figure 1—source data 1*). The eCLIP peaks for the RBPs used in the shRNA knockdown experiments are indicated and highlighted with arrows and red bars. The location of sgRNAs are shown with arrows (see *Figure 1—source data 3*), and the asterisks mark the sgRNAs from *Heskett et al., 2020*.

The online version of this article includes the following source data and figure supplement(s) for figure 1:

**Source data 1.** The location of significant peaks of eCLIP reads that map within ASAR and *XIST* genes.

**Source data 2.** eCLIP reads for 120 RBPs in 10 kb windows within ASARs.

**Source data 3.** Fosmids, BACs, and primers.

**Figure supplement 1.** CRISPR/Cas9 deletion of the ~7 kb RBPD.

**Figure supplement 1—source data 1.** Original image file of PCR products shown in *Figure 1—figure supplement 1C*.

both K562 and HepG2 cell lines (*Figure 1G*) and the abundant eCLIP reads from 35 different RBPs in 10 kb windows across the *ASAR6-141* lncRNA (*Figure 1C and D*; see *Figure 1—source data 2*), we were surprised to detect an ~7 kb region that contained virtually all of the significant eCLIP peaks in both cell lines (*Figure 1G*; and *Figure 1—source data 1*). The zoomed in view in *Figure 1G* shows alignment of 58 independent eCLIP peaks of RBP interactions within the ~7 kb RNA binding protein domain (RBPD). *Figure 1E* shows the 23 RBPs with the most abundant peaks of eCLIP reads within the ~7 kb RBPD. The eCLIP peaks associated with the RBPs chosen for shRNA knockdown experiments (see below) are highlighted in the zoomed in view in *Figure 1G*.

## Disruption of the ~7kb RBPD within *ASAR6-141* results in delayed replication

To determine if the ~7 kb RBPD within the *ASAR6-141* lncRNA contains replication timing activity, we used CRISPR/Cas9-mediated engineering to delete the ~7 kb sequence from the endogenous locus on human chromosome 6. For this analysis, we designed single guide RNAs (sgRNAs) to unique sequences on either side of the ~7 kb RBPD (see *Figure 1G*; and *Figure 1—figure supplement 1*). We expressed these sgRNAs in combination with Cas9 in human HTD114 cells and screened clones for deletion of the ~7 kb RBPD (see *Figure 1—figure supplement 1*). We chose HTD114 cells for this analysis because they maintain mono-allelic expression and asynchronous replication of both imprinted and random monoallelic genes, they have a stable karyotype that does not change significantly following transfection, drug selection and sub-cloning, and knockouts of the eight known ASARs, including *ASAR6-141*, result in DRT/DMC and chromosome structure instability in cis (*Donley et al., 2015*; *Heskett et al., 2020*; *Heskett et al., 2022*; *Stoffregen et al., 2011*; *Donley et al., 2013*; *Platt et al., 2018*; *Chang et al., 2007*; *Breger et al., 2005*). Because *ASAR6-141* expression is mono-allelic in the HTD114 cells (see *Figure 1A*; and *Heskett et al., 2020*), we isolated clones that had heterozygous or homozygous deletions of the ~7 kb RBPD. We determined which alleles were deleted based on retention of different base pairs of a heterozygous SNP located within the ~7 kb RBPD (see *Figure 1—figure supplement 1* and *Figure 1—source data 3*).

To assay replication timing of homologous chromosome pairs we quantified DNA synthesis in mitotic cells using a BrdU terminal label assay (*Figure 2A*; and *Smith et al., 2001*; *Smith and Thayer, 2012*). In addition, the HTD114 cells contain a centromeric polymorphism on chromosome 6, which allows for an unambiguous distinction between the two chromosome 6 homologs (*Stoffregen et al., 2011*; *Donley et al., 2013*; *Platt et al., 2018*; also see *Figure 2B*). For simplicity, we refer to the chromosome 6 with the larger centromere as CHR6A and to the chromosome 6 with the smaller centromere as CHR6B. From our previous studies we knew that the chromosome 6 with the smaller centromere is linked to the expressed allele of *ASAR6-141* [5]. *Figure 2B and C* shows the replication timing analysis on a mitotic cell where the ~7 kb RBPD was deleted from the expressed allele (CHR6B). Note that CHR6B contains more BrdU incorporation than CHR6A, indicating later replication of CHR6B (*Figure 2D*). Quantification of the BrdU incorporation in CHR6B and CHR6A in multiple cells indicated that deletion of the ~7 kb RBPD from the expressed allele resulted in a significant delay in replication timing (*Figure 2E*). This is in contrast to cells prior to deletion, or in cells containing a

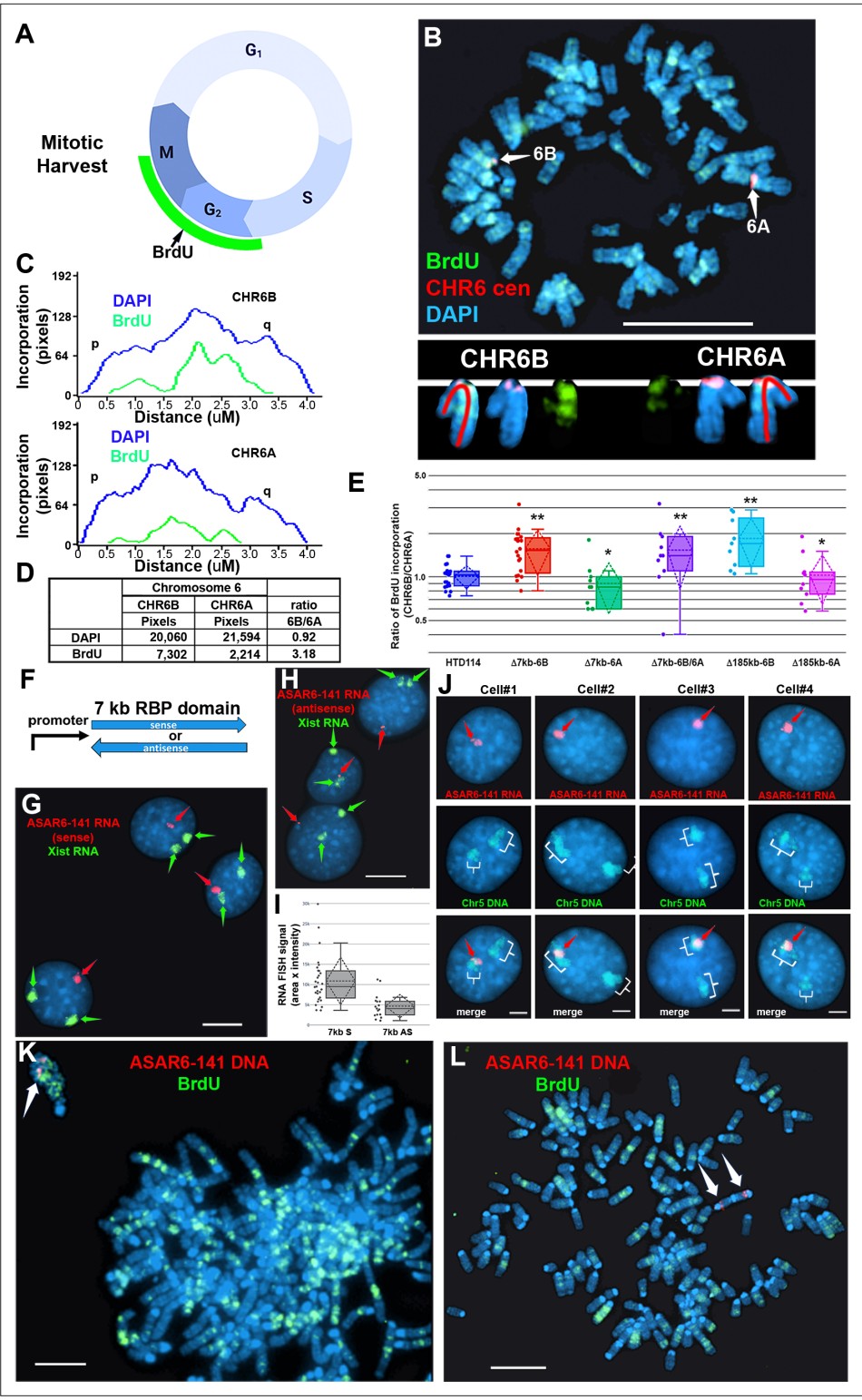

**Figure 2.** Replication timing in cells with *ASAR6–141~7* kb RBPD deletion or ectopic integration. (**A**) Schematic illustration of the BrdU terminal label protocol (*Smith and Thayer, 2012*). Cells were treated with BrdU (green) for 5 hr and then harvested for mitotic cells. (**B**) BrdU incorporation in HTD114 cells containing a heterozygous deletion of the ~7 kb RBPD. Cells containing a deletion of the ~7 kb RBPD from the expressed allele of *ASAR6-141* were exposed to BrdU, harvested for mitotic cells, and subjected to DNA FISH using a chromosome 6 centromeric probe (red; **B**). The larger centromere resides on the chromosome 6 with the silent *ASAR6-141* allele (CHR6A),

*Figure 2 continued on next page*

*Figure 2 continued*

and the smaller centromere resides on the chromosome 6 with the expressed *ASAR6-141* allele (CHR6B). DNA was stained with DAPI (blue). (**C and D**) DAPI staining and BrdU incorporation were quantified by calculating the number of pixels (area x intensity) and displayed as a ratio of BrdU incorporation in CHR6B divided by the BrdU incorporation in CHR6A. (**E**) Quantification of BrdU incorporation in multiple cells with heterozygous (Δ7kb-6B, expressed allele; or Δ7kb-6A, silent allele) or homozygous (Δ7kb-6B/6 A) deletions of the ~7 kb RBPD. Also shown is the quantification of BrdU incorporation in heterozygous deletions of the entire ~185 kb *ASAR6-141* gene from chromosome 6B (Δ185-6B) or 6 A (Δ185–6 A) (*Heskett et al., 2020*). Box plots indicate mean (solid line), standard deviation (dotted line), 25th, 75th percentile (box) and 5th and 95th percentile (whiskers) and individual cells (single points). p Values were calculated using the Kruskal-Wallis test (*Kruskal, 1964*). Values for a minimum of 10 individual cells are shown as dots. **p=0.00027 and *p=0.03. (**F**) Schematic view of the ~7 kb RBPD in both the sense and antisense orientation. The promoter used to drive expression is from *ASAR6*[9]. (**G and H**) Two color RNA FISH assay for expression of the ~7 kb RBPD transgenes. Individual clones were screened for RNA expressed from the *ASAR6-141* (~7 kb RBPD) transgenes (red, arrows), and RNA hybridization using an *Xist* probe (green, arrows) was used as positive control. (**G and H**) represent examples of cells containing the sense and antisense transgenes, respectively. (**I**) Quantitation of the size (pixels = area x intensity) of the RNA FISH signals expressed from the 7 kb sense (7 kb S) or antisense (7 kb AS) transgenes are shown. Box plots indicate mean (solid line), standard deviation (dotted line), 25th, 75th percentile (box) and 5th and 95th percentile (whiskers) and a minimul of 10 individual cells (single points) are shown. p Value of 0.006 was calculated using the Kruskal-Wallis test (*Kruskal, 1964*). Values for individual cells are shown as dots. (**J**) RNA from the ~7 kb RBPD remains localized to mouse chromosomes. RNA-DNA FISH using the ~7 kb RBPD as RNA FISH probe (red, arrows), plus a mouse chromosome 5 paint to detect chromosome 5 DNA (green, brackets). (**K**) Cells containing the sense ~7 kb RBPD transgene integrated into mouse chromosome 5 were exposed to BrdU (green), harvested for mitotic cells, and subjected to DNA FISH using the ~7 kb RBPD (red, arrows). Note that the chromosome that contains the transgene shows delayed mitotic condensation and more BrdU incorporation than any other chromosome within the same cell. (**L**) Cells containing the antisense ~7 kb RBPD transgene integrated into a mouse chromosome were exposed to BrdU (green), harvested for mitotic cells, and subjected to DNA FISH using the ~7 kb RBPD (red). Note that the chromosome that contains the transgene shows normal mitotic condensation and only a small amount of BrdU incorporation. Scale bars are 10 µM (**B, G, H**) 5 µM (**K, L**) and 2 µM (**J**).

The online version of this article includes the following figure supplement(s) for figure 2:

**Figure supplement 1.** Identification of mouse chromosomes containing the ~7 kb RBPD transgenes.

---

deletion of the ~7 kb RBPD from the silent allele (CHR6A), where the BrdU incorporation is comparable between CHR6A and CHR6B (*Figure 2E*). In addition, an asynchronous replication pattern was also present in cells with homozygous deletions of the 7 kb RBPD, with CHR6B representing the later replicating allele (*Figure 2E*). Furthermore, we found that the delayed replication associated with deletion of the ~7 kb RBPD on CHR6B is comparable to the delayed replication associated with deletion of the entire *ASAR6-141* locus from the same chromosome (i.e. CHR6B; *Figure 2E*; also see *Heskett et al., 2020*). These results indicate that the ~7 kb RBPD contains functional sequences for controlling replication timing, and further supports the conclusion that the expressed allele but not the silent allele controls replication timing of human chromosome 6 (*Heskett et al., 2020*).

## Ectopic integration of transgenes expressing the ~7kb RBPD into autosomes

To determine if the ~7 kb RBPD is sufficient to induce delayed replication in an ectopic integration assay, we generated two transgenes containing the ~7 kb RBPD, one in the sense orientation and one in the antisense orientation with respect to the promoter (*Figure 2F*). These transgenes were introduced into mouse cells and individual clones isolated. We initially screened individual clones for expression of the transgenes using RNA FISH with the ~7 kb RBPD as probe. Because the mouse cells that we used are female, we included an RNA FISH probe for *Xist* to serve as positive control for the RNA FISH assays. This two-color RNA FISH assay allowed us to identify clones with single sites of transgene expression, and allowed us to directly compare the RNA FISH signals from the transgenes to that of the endogenous *Xist* RNA expressed from the inactive X chromosome. *Figure 2* shows examples of the RNA FISH signals in cells expressing either the sense (*Figure 2G*) or antisense (*Figure 2H*) transgenes. Large clouds of RNA FISH hybridization signals that are comparable to *Xist* RNA hybridization signals were detected in 4 out of 12 clones expressing the sense transgene

(*Figure 2G*). In contrast, 0 out of 12 clones containing the antisense transgene expressed RNA FISH hybridization signals comparable to *Xist* RNA signals, but instead only small pinpoint sites of RNA FISH hybridization were detected (*Figure 2H*). Quantitation of the nuclear area occupied by the sense and antisense ~7 kb RNAs is shown in *Figure 2I*. Note that the mouse cells are tetraploid for the X chromosome and contain two *Xist* RNA hybridization signals, representing two inactive X chromosomes.

To identify the mouse chromosomes that contain the transgenes, we used DNA FISH using the ~7 kb RBPD DNA as probe in combination with the inverted DAPI banding pattern to identify the mouse chromosomes in metaphase spreads. We then confirmed the identity of the mouse chromosomes using DNA FISH on metaphase spreads using the ~7 kb RBPD in combination with BAC probes from the chromosomes of interest. For example, clone ~7 kb(+)A5, which contains the sense-strand transgene integrated in mouse chromosome 5 is shown in *Figure 2—figure supplement 1A*. Next, to determine if the RNA expressed from the ~7 kb RBPD sense strand transgene is retained within the chromosome territory that expresses the transgene, we used RNA-DNA FISH with the ~7 kb RBPD as RNA probe plus a chromosome paint to detect mouse chromosome 5 DNA in clone ~7 kb(+)A5. *Figure 2J* shows that the ~7 kb RBPD RNA is localized within one of the chromosome 5 DNA hybridization signals, indicating that the ~7 kb RBPD RNA is retained within the chromosome 5 territory.

Next to determine if the ~7 kb RBPD transgene alters replication timing of mouse chromosome 5, we visualized DNA synthesis in mitotic cells using the BrdU terminal label assay (see *Figure 2A*; and *Smith et al., 2001*; *Smith and Thayer, 2012*). Cells were exposed to BrdU for 5 hr, harvested for mitotic cells, and processed for BrdU incorporation and for DNA FISH using the ~7 kb RBPD transgene to identify the integrated chromosome. *Figure 2K* shows an example of this replication-timing assay in a mitotic cell containing the ~7 kb RBPD sense strand transgene integrated into mouse chromosome 5. Note that the chromosome containing the ~7 kb RBPD transgene is delayed in mitotic condensation and contains more BrdU incorporation than any other chromosome within the same cell. In contrast, integration of the transgene with the ~7 kb RBPD in the antisense orientation did not result in delayed condensation nor in delayed replication of mouse chromosomes (*Figure 2L*).

## Depletion of RBPs disrupts the chromosome territory localization of ASAR lncRNAs

The results described above identified 99 RBPs expressed in two different cell lines that interact with four different ASAR lncRNAs (see *Figure 1—source data 1*). Next, to determine if the ASAR associated RBPs function in the chromosome territory localization of ASAR lncRNAs, we analyzed ASAR lncRNA localization in cells with shRNA depletion of 14 different RBPs. *Figure 3A–J* shows examples of antibody staining for HNRNPU, HNRNPUL1, HNRNPC, HNRNPM, and HLTF in K562 cells expressing either empty vector or vectors expressing shRNAs to individual RBPs. Quantitation of the immunofluorescence for each RBP in empty vector versus shRNA expressing cells, indicated a significant depletion of each RBP (*Figure 3K*; also see *Figure 3—figure supplement 1* for western blots).

Next, to determine if the nuclear localization of *ASAR6-141* lncRNA was affected by RBP depletions, we used RNA FISH assays in K562 cells expressing shRNAs against the 14 RBPs. For this analysis, we used RNA FISH to detected *ASAR6-141* lncRNA, and because K562 cells are female we also included a probe for *XIST* RNA as positive control for the RNA FISH. *Figure 3L* shows the expected RNA FISH patterns for both *ASAR6-141* and *XIST* lncRNAs in cells expressing empty vector. In contrast, cells expressing shRNAs directed at *HNRNPU*, *HNRNPUL1*, *HNRNPC*, *HNRNPM*, H*LTF*, *HNRNPA1*, *HNRNPL*, *KHSRP*, or *UCHL5* showed a nearly complete absence of the highly localized nuclear RNA hybridization signals for *ASAR6-141* lncRNA, and the appearance of large punctate cytoplasmic hybridization signals (*Figure 3M–U*). We also noticed an overall increase in a diffuse *ASAR6-141* lncRNA nuclear and cytoplasmic hybridization signals. In contrast, cells expressing shRNAs directed at *PTBP1*, *PTBP2*, or *MATR3* showed only the typical nuclear localized hybridization signals for both *ASAR6-141* and *XIST* lncRNAs (*Figure 3V*; and *Figure 3—figure supplement 2A–D*). In addition, localized nuclear *XIST* RNA hybridization signals were detected in all shRNA treated cells (*Figure 3M–V* and *Figure 3—figure supplement 2A–D*). However, we detected cytoplasmic foci of *XIST* RNA hybridization that colocalized with the *ASAR6-141* lncRNA, and a significant decrease in the size of the nuclear *XIST* RNA hybridization signals (*Figure 3W*). These observations indicate that at least some of the *XIST* RNA was dissociated from the inactive X chromosome and colocalized with *ASAR6-141* lncRNA in large cytoplasmic foci.

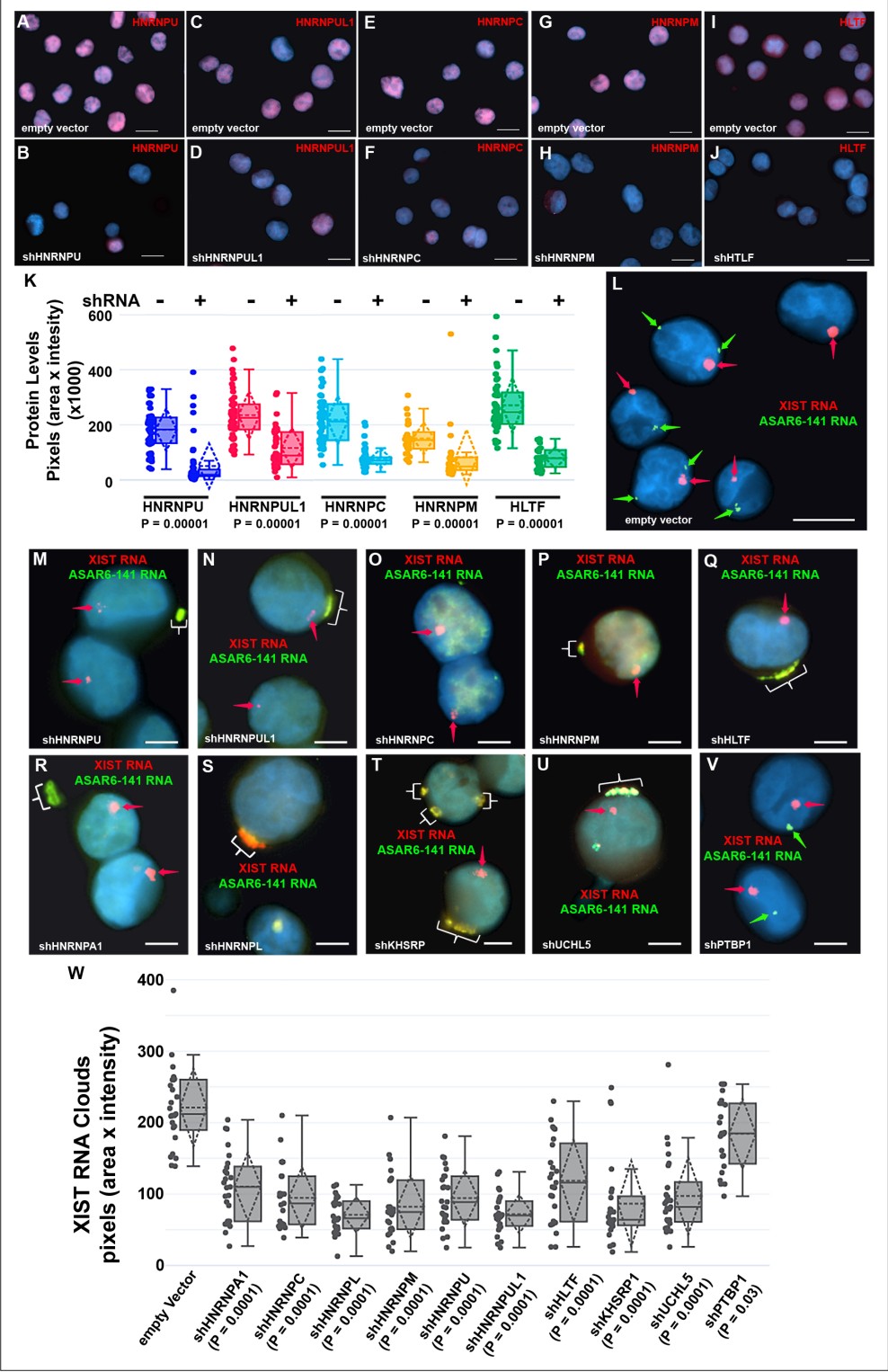

**Figure 3.** Depletion of RBPs results in disruption of the chromosome territory localization of *ASAR6-141* RNA. (**A–J**) shRNA mediated depletion of RBPs. K562 cells were transfected with empty vector (**A, C, E, G, and I**) or vectors expressing shRNAs directed against *HNRNPU* (**B**), *HNRNPUL1* (**D**), *HNRNPC* (**F**), *HNRNPM* (**H**), or *HLTF* (**J**). (**K**) Cells were stained with the appropriate antibodies and quantitation of each RBP was determined in >25 individual cells. Box plots indicate mean (solid line), standard deviation (dotted line), 25th, 75th percentile (box), 5th and 95th percentile (whiskers), and individual cells (single points) are indicated. p Values were calculated using the Kruskal-Wallis test (**Kruskal, 1964**). (**L–V**) RNA-DNA FISH for *ASAR6-141* (green) and XIST (red) RNA. K562 cells were

*Figure 3 continued on next page*

*Figure 3 continued*

transfected with empty vector (**L**) or vectors expressing shRNAs against *HNRNPU* (**M**), *HNRNPUL1* (**N**), *HNRNPC* (**O**), *HNRNPM* (**P**), HLTF (**Q**), *HNRNPA1* (**R**), *HNRNPL* (**S**), *KHSRP* (**T**), *UCHL5* (**U**), or *PTBP1* (**V**). The red arrows mark the RNA FISH signals for XIST. The brackets mark cytoplasmic regions that hybridized to both RNA FISH probes. DNA was stained with DAPI, and Bars are 10 μM (**A–J and L**) or 5 *u*M (**M–V**).(**W**) Quantitation of the *XIST* RNA FISH signals. K562 cells transfected and processed for RNA FISH as in L-V were analyzed for quantitation of *XIST* RNA cloud size (pixels: area X intensity) in >25 individual cells. Box plots indicate mean (solid line), standard deviation (dotted line), 25th, 75th percentile (box), 5th and 95th percentile (whiskers) and individual cells (single points). p Values were calculated using the Kruskal-Wallis test (**Kruskal, 1964**).

The online version of this article includes the following source data and figure supplement(s) for figure 3:

**Figure supplement 1.** RBP protein levels in cells expressing shRNAs.

**Figure supplement 1—source data 1.** Original image files for the western blots.

**Figure supplement 2.** RNA FISH on cells transfected with shRNAs that did not disrupt ASAR localization.

## Ectopic integration of the ~7kb RBPD transgene into the inactive X chromosome

The results described above indicate that shRNA depletion of nine different RBPs resulted in dissociation of *ASAR6-141* lncRNA from its parent chromosome, and resulted in a decreased nuclear *XIST* RNA hybridization signal with concomitant colocalization of both RNAs in cytoplasmic foci (see *Figure 3*), suggesting that *ASAR6-141* and *XIST* lncRNAs are retained on their respective chromosome territories via similar mechanisms, which is supported by the large overlap in RBPs associated with both RNAs (*Figure 1E and F*; and *Figure 1—source data 1*). In addition, ectopic integration of *Xist* transgenes into autosomes has been an instrumental tool in characterization of *Xist* functions, including the ability to delay replication timing and induce gene silencing (*Minks and Brown, 2009*; *Pintacuda et al., 2017*; *Lee et al., 1996*). Therefore, we next sought to determine the consequences of ectopic integration of the *ASAR6−141*~7 kb RBPD transgene into the inactive X chromosome. The ectopic integration assays described above involved random integration of transgenes followed by an RNA FISH screen for clones expressing the ~7 kb RBPD, which included an RNA FISH probe for *Xist* RNA as positive control. Therefore, to identify transgene integrations into the inactive X chromosome, we simply screened an additional 50 clones for colocalization of RNA FISH hybridization signals for both *Xist* and the ~7 kb RBPD transgene. *Figure 4A* shows this RNA FISH assay on cells where the *Xist* and ~7 kb RBPD hybridization signals are colocalized. We note that the intensity of the *Xist* RNA FISH signal was enhanced in the region where the ~7 kb RBPD hybridization signal was detected (*Figure 4B*), suggesting that the presence of the ~7 kb RBPD RNA promoted preferential localization of *Xist* RNA in the same region of the X chromosome territory that contains the ~7 kb RBPD RNA.

To confirm that the mouse X chromosome contains the transgene, we used DNA FISH using the ~7 kb RBPD DNA as probe in combination with an X chromosome paint probe on metaphase spreads. An example of this analysis is shown in *Figure 2—figure supplement 1B–C*, and indicates that the ~7 kb RBPD transgene is integrated into an X chromosome. During this analysis, we detected DMC on X chromosomes, and the X chromosomes with DMC contain the ~7 kb RBPD transgene (*Figure 4C and D*).

Next, to determine if the ~7 kb RBPD transgene alters replication timing of the inactive X chromosome, we visualized DNA synthesis in mitotic cells using our BrdU terminal label assay (see *Figure 2A*; and *Smith et al., 2001*; *Smith and Thayer, 2012*). Cells were exposed to BrdU for 5 hr, harvested for mitotic cells, and processed for BrdU incorporation and for FISH using the ~7 kb RBPD transgene plus an X chromosome paint probe to identify the integrated chromosome. *Figure 4E–H* shows an example of delayed replication and delayed mitotic condensation of the X chromosome in a mitotic cell containing the ~7 kb RBPD sense transgene integrated into the inactive X chromosome. We conclude that integration of the 7 kb RBPD transgene is sufficient to cause DRT/DMC on the inactive X chromosome.

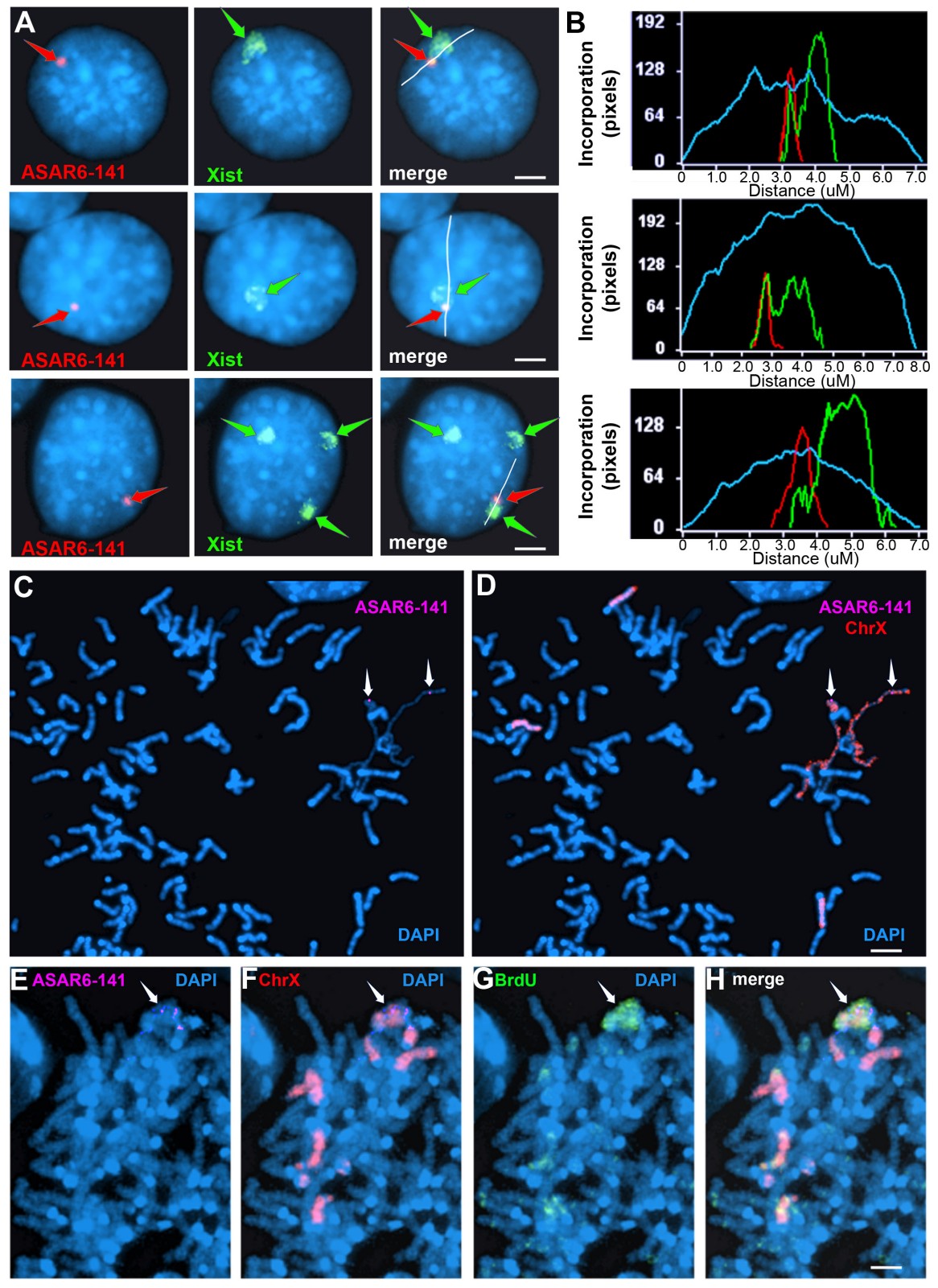

**Figure 4.** Ectopic integration of the *ASAR6−141~7 kb RBPD* transgene into an inactive X chromosome. (**A**) Two color RNA FISH assay for expression of the sense ~7 kb RBPD transgene plus *Xist* RNA. Examples of three different cells showing colocalization of the ASAR RNA and *Xist* RNA. Individual cells were processed for expression of the sense strand ~7 kb RBPD transgene (left panels, red, arrow) in combination with RNA FISH for *Xist* (middle panels, green, arrow). The right panels show the merged images. The white lines show the path for the pixel intensity profiles used for panel B. (**B**) Pixel

*Figure 4 continued on next page*

*Figure 4 continued*

intensity profiles across the *Xist* RNA hybridization domain showing enhanced signal intensity over the ~7 kb RBPD RNA hybridization domain. (**C and D**) Delayed mitotic chromosome condensation of the X chromosome that contains the ~7 kb RBPD transgene. An example of a mitotic spread processed for DNA FISH using the ~7 kb RBPD as probe (magenta, panel C) plus a mouse X chromosome paint (red, panel D). The arrows mark the location of the transgene hybridization signal. (**E–H**) Cells containing the ~7 kb RBPD transgene integrated into the mouse X chromosome were exposed to BrdU (G, green), harvested for mitotic cells, and subjected to DNA FISH using the ~7 kb RBPD (E, magenta) plus a mouse X chromosome paint (F, red). The merged images are shown in panel H. Scale bars are 2 μM.

## Disruption of the chromosome associated localization of *ASAR6* and *ASAR1-187* lncRNAs

To determine if the chromosomal localization of additional ASAR RNAs was also affected by RBP knockdowns, we carried out RNA-DNA FISH for *ASAR1-187* and *ASAR6* lncRNAs. For this analysis, we used Fosmid probes to detect the ASAR RNAs plus BAC probes to detect the DNA near the ASAR loci located on chromosomes 1 and 6 (see *Figure 1—source data 3*). *Figure 5A–C* shows this RNA FISH assay on K562 cells expressing empty vector, and indicates that the RNA FISH signals for both ASARs are closely associated with the DNA FISH signals on their respective chromosomes, that is *ASAR1-187* RNA shows close association with the chromosome 1 BAC DNA, and *ASAR6* RNA shows close association with the chromosome 6 BAC DNA. In contrast, cells with shRNA depletion of HNRNPA1, HNRNPC, HNRNPL, HNRNPM, HNRNPU, HLTF, KHSRP, or UCHL5 showed a dispersed punctate nuclear pattern for both ASAR RNAs (*Figure 5D–K*). In addition, large punctate cytoplasmic foci were detected, with the two ASAR RNA hybridization signals colocalized.

Next, to determine if depletion of RBPs would result in loss of the ASAR RNA chromosome localization in a second cell type, we carried out RNA-DNA FISH assays on shRNA depleted HTD114 cells. We chose HTD114 cells because we previously used these cells to detect the chromosome territory localization of all eight ASAR RNAs and to measure the synchronous replication timing of five autosome pairs both before and after ASAR disruptions (*Donley et al., 2015*; *Heskett et al., 2020*; *Heskett et al., 2022*; *Stoffregen et al., 2011*; *Donley et al., 2013*; *Breger et al., 2005*; and see *Figure 2*). *Figure 5L–N* shows the RNA-DNA FISH assay in HTD114 cells expressing empty vector. We again note the tight association of the ASAR RNAs with the genomic loci where they are transcribed. In contrast, cells expressing shRNA against *HNRNPU* show loss of the chromosome associated RNA hybridization signals for both ASAR RNAs, and instead show diffuse nuclear and cytoplasmic hybridization, with the presence of large punctate cytoplasmic foci that hybridize to both ASAR RNA FISH probes (*Figure 5O–Q*). Similar results were obtained following shRNA depletion of HNRNPA1, HNRNPC, HNRNPL, HNRNPM, HNRNPUL1, KHSRP, HLTF, or UCHL5 (*Figure 5—figure supplement 1*). In contrast, expression of shRNAs against *SAFB*, *SAFB2*, *SAFB* plus *SAFB2*, *CTCF*, or *MATR3* did not alter the *ASAR1-187* or *ASAR6* RNA nuclear hybridization signals in HTD114 cells (*Figure 3—figure supplement 2E–J*).

## Depletion of RBPs causes asynchronous replication on all autosome pairs

Next, to determine if ASAR associated RBPs function during the normally synchronous replication timing program that occurs on pairs of homologous chromosomes, we analyzed DNA synthesis in HTD114 cells depleted for the 9 RBPs implicated in ASAR RNA localization (HNRNPU, HNRNPA1, HNRNPC, HNRNPL, HNRNPM, HNRNPUL1, KHSRP, HLTF, or UCHL5) and for five RBPs that showed no effect on ASAR RNA localization (MATR3, PTBP1, PTBP2, SAFB, or SAFB2). Using the BrdU terminal label assay (*Smith et al., 2001*; *Smith and Thayer, 2012*) in cells expressing the empty vector, we found that each autosome pair displays a stereotypical and synchronous BrdU incorporation pattern (*Figure 6A*). In contrast, following HNRNPU depletion, individual mitotic spreads displayed dramatic differential incorporation of BrdU into pairs of homologous chromosomes, examples of asynchronous replication timing between every pair of autosomes are shown in *Figure 6B*. Note that each chromosome pair contains dramatic differential BrdU incorporation into chromosome homologs, which represents considerable asynchronous replication timing. *Figure 6C and D* show examples for chromosome pairs 4 and 18 (also see *Figure 6—figure supplement 1C*). Quantification of the BrdU incorporation in chromosomes 1 and 6 (see *Figure 6—figure supplement 1*) in multiple cells depleted for HNRNPU, HNRNPUL1, HLTF, KHSRP, UCHL5, or HNRNPA1 indicated that both pairs of chromosomes

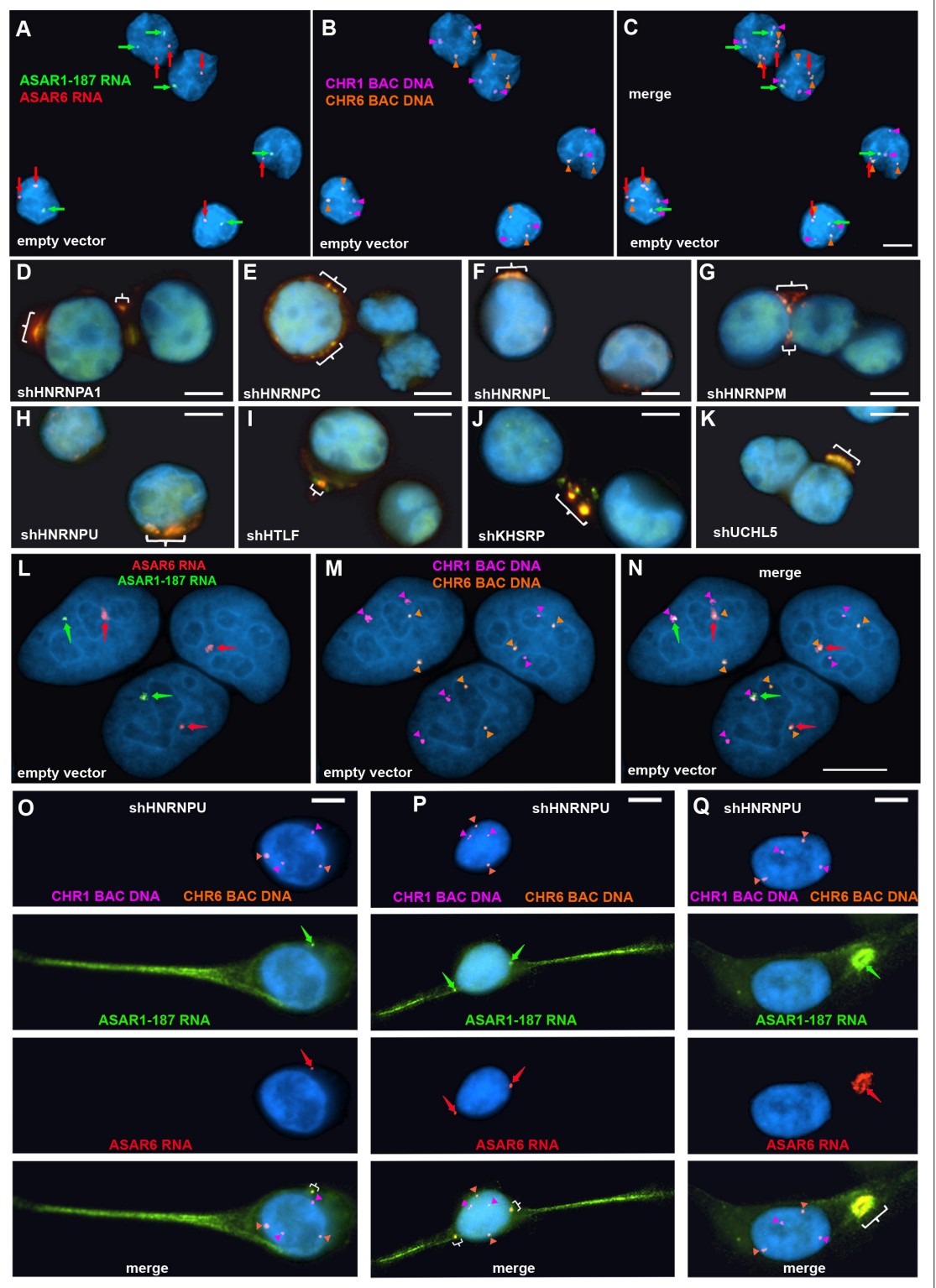

**Figure 5.** Depletion of RBPs results in disruption of the chromosome territory localization of ASAR RNAs. shRNA-mediated depletion of RBPs in K562 cells (**A–K**) or HTD114 cells (**L–Q**). Cells were transfected with empty vector (**A–C and L–N**) or vectors expressing shRNAs directed against *HNRNPA1* (**D**), *HNRNPC* (**E**), *HNRNPL* (**F**), *HNRNPM* (**G**), *HNRNPU* (H, and O-Q), *HLTF* (**I**), *KHSRP* (**J**), or *UCHL5* (**K**). Cells were processed for RNA FISH with probes for *ASAR6* (red, arrows) and *ASAR1-187* (green, arrows). Cells were also processed for DNA FISH using BAC DNA to detect chromosome 1 (CHR1 BAC DNA, magenta) and chromosome 6 (CHR6 BAC DNA, orange; **A–C and L–Q**). Arrows mark the sites of RNA FISH hybridization and arrowheads mark the sites of DNA hybridization. RNA-DNA FISH for *ASAR1-187* (RNA in green, arrows; DNA in magenta and arrowheads), *ASAR6* (RNA in red, arrows;

*Figure 5 continued on next page*

*Figure 5 continued*

DNA in orange and arrowheads). The brackets mark the cytoplasmic regions that hybridized to both RNA FISH probes. DNA was stained with DAPI, and scale bars are 10 µM.

The online version of this article includes the following figure supplement(s) for figure 5:

**Figure supplement 1.** Disruption of the chromosome territory localization of *ASAR1-187* and *ASAR6* RNAs following shRNA depletion of RBPs.

display significant asynchrony (***Figure 6E and F***). In addition, the DRT/DMC phenotype was detected in ~30% of mitotic spreads following shRNA-mediated depletions (two examples are shown in ***Figure 6G and H***; also see ***Figure 6—figure supplement 2***). Furthermore, shRNA-mediated depletion of HNRNPC, HNRNPL and HNRNPM resulted in dramatic alterations in chromosome morphology that included asynchronous replication and abnormal mitotic condensation, including ~10% of mitotic spreads with one or more chromosomes that displayed the DRT/DMC phenotype (***Figure 6—figure supplement 2***).

## C0T-1 DNA hybridizes to *ASAR6* RNA

Finally, to determine the relationship between the RNA species detected by C0T-1 DNA and ASAR RNAs in RNA FISH assays, we used human C0T-1 DNA as an RNA FISH probe on mouse cells expressing an *ASAR6* BAC transgene (***Donley et al., 2013***). Because human C0T-1 DNA does not cross hybridize with mouse RNA or DNA in FISH assays (***Hall et al., 2014***), we were able to detect the chromosome territory localized expression of human *ASAR6* RNA expressed from the transgene that is integrated into one of the mouse chromosome 3 homologs using human C0T-1 DNA as RNA FISH probe (***Figure 7A and B***), indicating that C0T-1 DNA detects *ASAR6* RNA.

## Discussion

We recently identified 68 ASAR candidates, and genetically validated five out of five as controlling replication timing of individual human autosomes, bringing the total of genetically validated ASARs to eight (***Donley et al., 2015***; ***Heskett et al., 2020***; ***Heskett et al., 2022***; ***Stoffregen et al., 2011***). These eight ASARs (*ASAR1-187*, *ASAR6*, *ASAR6-141*, *ASAR8-2.7*, *ASAR9-23*, *ASAR9-24*, *ASAR9-30*, and *ASAR15*) display striking epigenetically controlled differential allelic expression and replication timing, and their RNAs remain localized to their parent chromosome territories (***Donley et al., 2015***; ***Heskett et al., 2020***; ***Heskett et al., 2022***; ***Stoffregen et al., 2011***). In this report, we used publicly available RNA-protein interaction data from ENCODE to identify RBPs that interact with multiple ASAR lncRNAs. One unanticipated observation that came from this analysis was that an ~7 kb domain within the ~185 kb *ASAR6-141* lncRNA contained virtually all of the significant peaks of RBP interactions in two different cell lines. Using genetic deletion and ectopic integration assays we found that this ~7 kb RBPD contains functional sequences for controlling replication timing of entire chromosomes in cis. We also found that shRNA-mediated depletion of 9 different ASAR-associated RBPs results in loss of the chromosome territory association of multiple ASAR RNAs. In addition, we found that depletion of HNRNPA1, HNRNPC, HNRNPL, HNRNPM, HNRNPU, HNRNPUL1, HLTF, KHSRP, or UCHL5 alters the normally synchronous replication timing that occurs between pairs of homologous chromosomes resulting in the DRT/DMC phenotype, recapitulating the effect of individual ASAR knockouts on a genome-wide scale.

Previous studies have suggested that C0T-1 RNA plays a dynamic structural role that promotes the open architecture of active chromosome territories (***Creamer et al., 2021***; ***Hall et al., 2014***). C0T-1 RNA is predominantly L1 sequences, and these L1 RNAs remain associated with the chromosome territories where they are transcribed (***Hall et al., 2014***). Additional studies have proposed that repRNA is an important regulator of the dynamic organization of genomic loci into membrane-less subcompartments with distinct nuclear functions (reviewed in ***Frank and Rippe, 2020***). Other studies have proposed that HNRNPU nonspecifically interacts with 'RNA debris' to create a dynamic nuclear mesh that regulates interphase chromatin structure (***Nozawa and Gilbert, 2019***; ***Nozawa et al., 2017***). One important distinction that can be made between ASAR RNAs and these other hnRNAs (i.e. C0T-1 RNA, repRNA, caRNA, and 'RNA debris' ***Nozawa and Gilbert, 2019***; ***Creamer et al., 2021***; ***Nozawa et al., 2017***; ***Frank and Rippe, 2020***; ***Hall et al., 2014***; ***Kolpa et al., 2022***) is that ASARs have been defined genetically, and these other RNAs have not. Thus, we have used both loss-of-function and

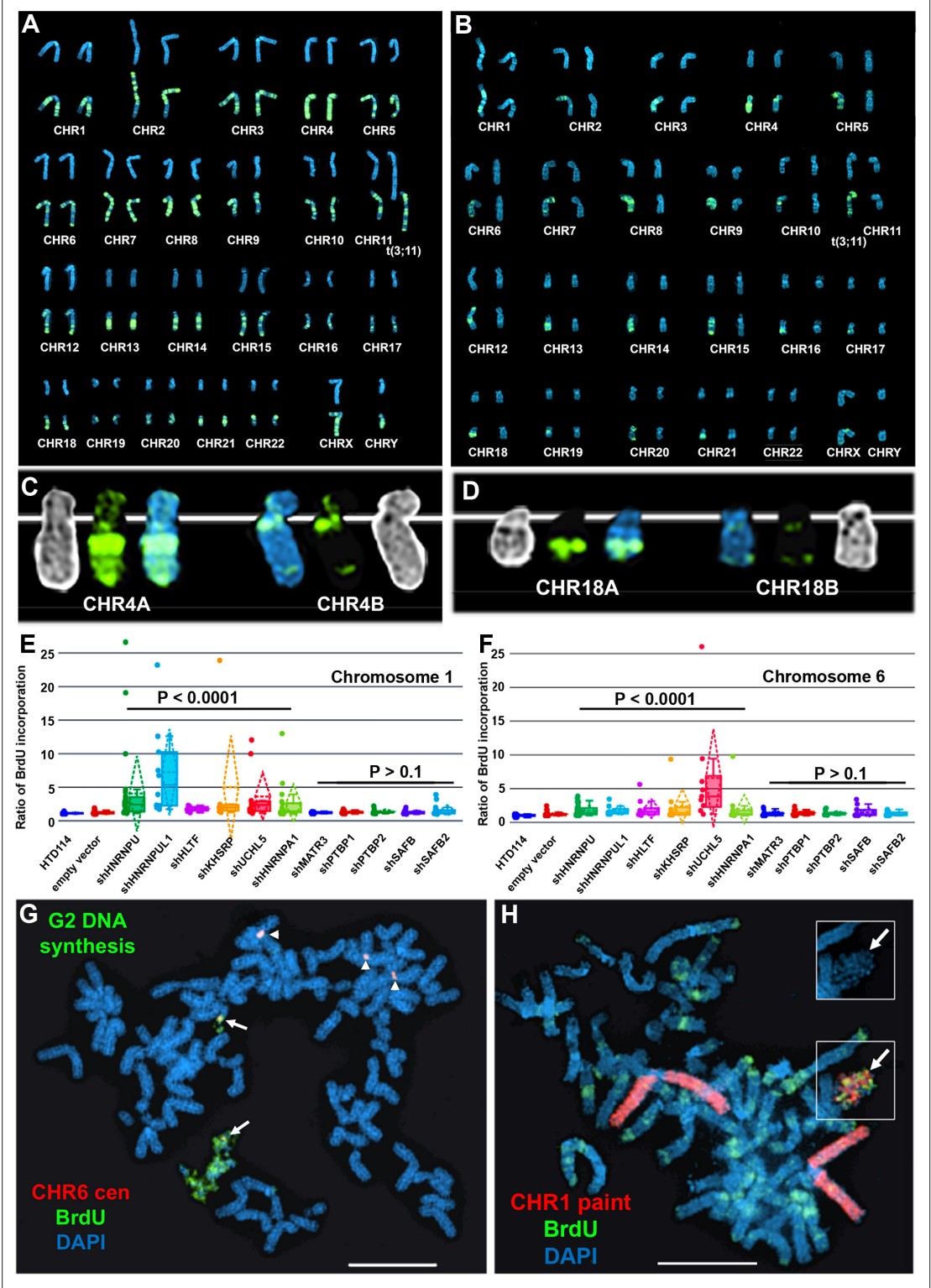

**Figure 6.** Depletion of RBPs results in asynchronous replication on autosome pairs. (**A**) BrdU incorporation in parental HTD114 cells. Cells were exposed to BrdU for five hours and processed for BrdU incorporation using an antibody against BrdU. This panel represents chromosomes from multiple mitotic spreads showing representative BrdU incorporation in chromosome pairs. Both homologs of autosome pair were captured from the same mitotic cell, and each pair displays a typical BrdU incorporation pattern that is consistent with synchronous replication timing. (**B**) HTD114 cells were transfected with the *HNRNPU* shRNA expression vector, exposed to BrdU for five hours, and processed for BrdU incorporation. This panel represents chromosomes from multiple mitotic spreads showing representative BrdU incorporation in pairs of autosomes. Both homologs of autosome pairs were captured from the

*Figure 6 continued on next page*

*Figure 6 continued*

same mitotic cell, and each pair displays a differential BrdU incorporation pattern that is consistent with asynchronous replication timing. (**C**) Shows two chromosome 4 homologs (CHR4A and CHR4B) side by side. (**D**) Shows the BrdU incorporation in chromosome 18 homologs (CHR18A and CHR18B). For each chromosome pair the inverted DAPI staining (black and white), BrdU incorporation (green), and DAPI staining (blue) are shown. (**E and F**) Quantification of BrdU incorporation in multiple cells depleted for HNRNPU, HNRNPUL1, HTLF, KHSRP, UCHL5, HNRNPA1, MATR3, PTBP1, PTBP2, SAFB, and SAFB2. Box plots indicate mean (solid line), standard deviation (dotted line), 25th, 75th percentile (box), 5th and 95th percentile (whiskers) and a minimum of 10 individual cells (single points) are indicated. p Values were calculated using the Kruskal-Wallis test (*Kruskal, 1964*). (**G**) DRT/DMC on chromosome 6 following HNRNPU depletion. HTD114 cells were transfected with the *HNRNPU* shRNA expression vector, exposed to BrdU for 5 hr, harvested for mitotic cells, and processed for BrdU incorporation (green) and DNA FISH using a chromosome 6 centromeric probe (red). (**H**) DRT/DMC on chromosome 1 following UCHL5 depletion. HTD114 cells were transfected with the *UCHL5* shRNA expression vector, exposed to BrdU for 5 hr, harvested for mitotic cells, and processed for BrdU incorporation (green) and DNA FISH using a chromosome 1 paint as probe (red). The arrow marks the chromosome 1 with DRT/DMC, and the inset shows only the DAPI staining of the chromosome 1 highlighting the DMC.

The online version of this article includes the following figure supplement(s) for figure 6:

**Figure supplement 1.** Replication timing in cells transfected with shRNA expression vectors.

**Figure supplement 2.** Replication timing alterations in shRNA depleted cells.

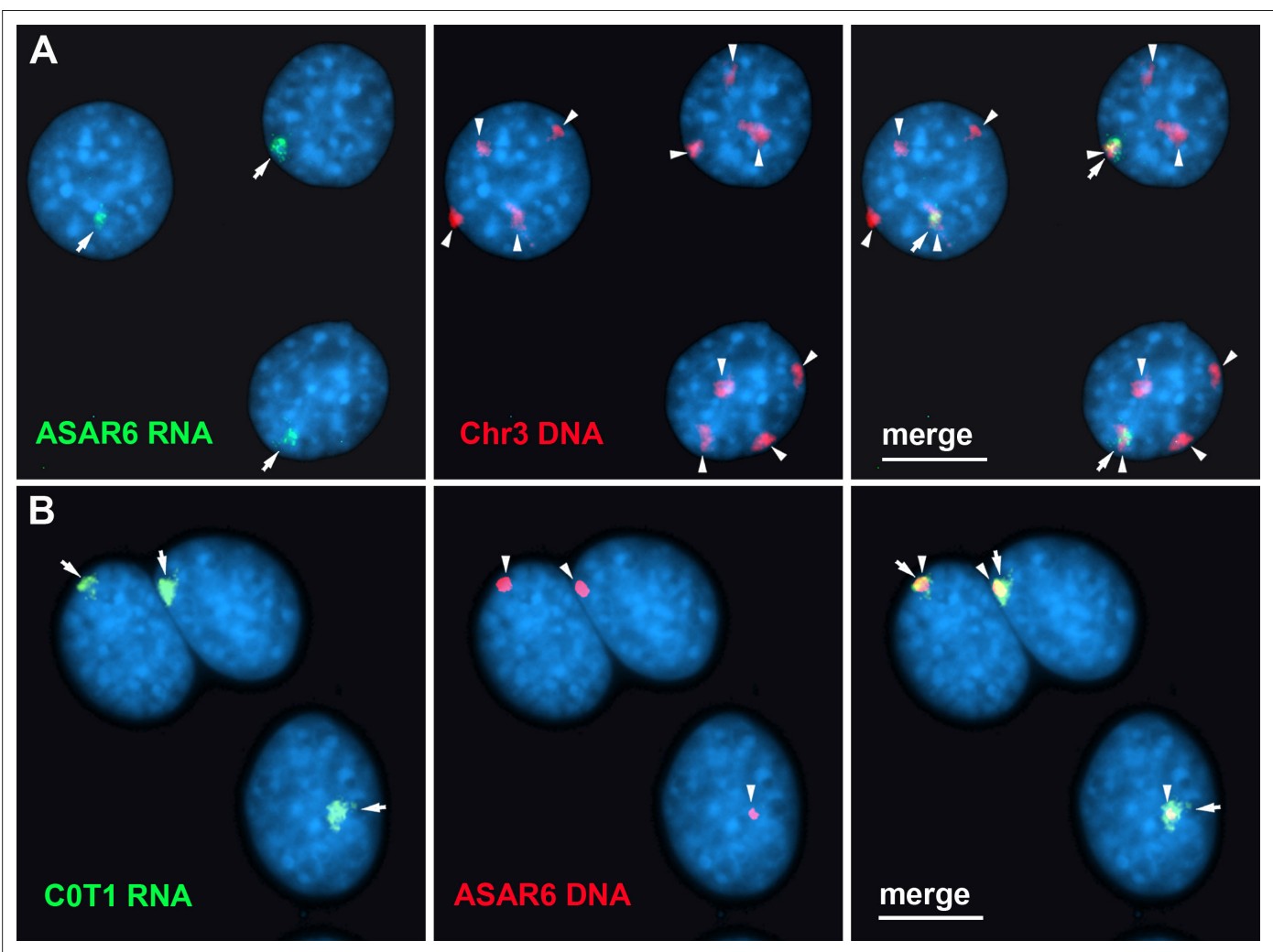

**Figure 7.** Human C0T-1 DNA detects *ASAR6* RNA. (**A and B**) RNA-DNA FISH on mouse cells containing an *ASAR6* BAC transgene integrated into mouse chromosome 3 *Donley et al., 2013*. (**A**) RNA-DNA FISH using an *ASAR6* fosmid probe to detect *ASAR6* RNA (green; arrows) plus a mouse chromosome 3 paint probe to detect DNA (red; arrowheads). (**B**) RNA-DNA FISH using human C0T-1 DNA to detect RNA (green; arrows) plus the *ASAR6* BAC (red arrowheads) to detect DNA.

gain-of-function assays to define the roles that ASARs play in chromosome dynamics, which includes control of chromosome-wide replication timing and mitotic chromosome condensation (*Donley et al., 2015*; *Heskett et al., 2020*; *Heskett et al., 2022*; *Stoffregen et al., 2011*; *Donley et al., 2013*). We also found that C0T-1 DNA, when used as an RNA FISH probe, can detect *ASAR6* RNA that is expressed and localized to an individual chromosome territory, indicating that at least some of the RNA FISH signal detected by C0T-1 DNA represents ASAR RNA. Given our recent discovery of ASAR RNAs expressed from every autosome, and encoded by ~3% of the human genome (*Heskett et al., 2022*), we propose that ASAR RNAs represent functional 'chromosome associated RNA' species that control chromosome dynamics within mammalian nuclei. Whether or not there are nuclear functions of other chromosome associated hnRNAs that are independent of ASARs will require genetic and/or functional validation.

Here we identified several hnRNPs (HNRNPA1, HNRNPC, HNRNPL, HNRNPM, HNRNPU, and HNRNPUL1) as important interaction partners for the chromosome territory localization of multiple ASAR RNAs. In addition, we found that the SWI/SNF related transcription factor HLTF, which was recently shown to regulate replication fork reversal, prevent alternative mechanisms of stress-resistant DNA replication, and mediates cellular resistance to replication stress (*Bai et al., 2020*,) is required for the chromosome territory localization of multiple ASAR RNAs and for the synchronous replication of every autosome pair. Furthermore, we found that the KH domain (HNRNPK Homology domain) containing protein KHSRP, and the ubiquitin hydrolase UCHL5 also contribute to the chromosome territory localization of ASAR RNAs. We found that depletion of HNRNPA1, HNRNPC, HNRNPL, HNRNPM, HNRNPU, HNRNPUL1, HLTF, KHSRP, or UCHL5 proteins resulted in a dramatic genome-wide disruption of the normally synchronous replication timing program on all autosome pairs. Taken together our results indicate that the association of ASAR RNAs with multiple hnRNPs in combination with other nuclear RBPs is an essential step in the regulation of the synchronous replication timing program on mammalian chromosomes.

While these studies have demonstrated a role for multiple RBPs in genome-wide chromosome replication timing, one limitation of these studies is that we only tested 14 out of the ~100 RBPs shown by eCLIP to associate with multiple ASAR RNAs. It is likely that additional insights into the network of RBP/ASAR interactions could be gained by extending our replication timing analyses to knockdowns of additional RBPs. Another limitation of these studies is that all of the RBPs implicated in control of replication timing are known to play various roles in other essential nucleic acid metabolic pathways, including transcription, splicing, mRNA export, and RNA degradation (*Geuens et al., 2016*; *Domanski et al., 2022*.) It is possible that knocking down these RBPs has an indirect effect on replication related to these other functions of the RBPs, rather than a direct effect on ASAR function. However, the RBP knockdowns that disrupt genome-wide synchronous replication also disrupt chromosome association of multiple ASAR RNAs. Our previous work demonstrated that the ability of an ectopically integrated *ASAR6* transgene to cause delayed replication timing of a mouse chromosome required the ASAR6 RNA (*Platt et al., 2018*.) Hence, we expect that displacement of ASAR RNA from chromosome territories upon RBP knockdown is a primary cause of the genome-wide replication timing asynchrony, regardless of whether the displacement of ASAR RNA from the chromosome is a direct or indirect effect of RBP knockdown. A final limitation of these studies is that the precise mechanism by which ASARs and RBPs interact to control replication timing remains unknown. One intriguing possibility that is consistent with our data is that the ASAR RNAs serve as scaffolds for the assembly of extensive RNA-protein and protein-protein complexes that form a chromosome-tethered liquid-liquid phase that facilitates the recruitment of the requisite chromatin modifiers and DNA replication machinery. In support of this notion, HNRNPA1 and HNRNPC are components of the 40 S hnRNP particle, which forms a nuclear biomolecular condensate (*Domanski et al., 2022*). In addition, HNRNPU and HNRNPUL1 interact with each of the 40 S hnRNP particle subunits (*Britton et al., 2014*), suggesting the possibility of extensive protein-protein interactions. ASAR RNAs normally remain associated exclusively with the chromosome territory from which they are transcribed, and ASAR RNAs on different chromosomes remain separate. RBP knockdowns that disrupt synchronous replication timing also release ASAR RNAs from their chromosome territories, allowing colocalization of ASAR RNAs from different chromosomes (i.e. *ASAR6* and *ASAR1-187*, see *Figure 5O–Q*), often in nuclear and/or cytoplasmic foci reminiscent of the mixing of liquid-liquid phases. Additional support for this model is the observation that Xist RNA participates in the assembly of a nuclear heteromeric

condensate essential for gene silencing on the inactive X chromosome (*Pandya-Jones et al., 2020*). It is worth noting that ectopic integration of Xist transgenes has been a useful assay for characterization of Xist functions, including the ability to delay replication timing and induce gene silencing on autosomes (reviewed *Minks and Brown, 2009*). Thus, our observation that the ~7 kb RBPD RNA induces DRT/DMC on the inactive X chromosome, and associates with the *Xist* RNA cloud (*Figure 4A and B*), suggests that the ~7 kb RBPD RNA interferes with Xist RNA function within the heteromeric condensate. One intriguing observation, which is largely ignored by the X inactivation field, is that deletion of the Xist gene on either the active or inactive X chromosomes results in delayed replication timing of the X chromosome (*Diaz-Perez et al., 2005*; *Diaz-Perez et al., 2006*). Thus, loss of function mutations of either ASARs or *Xist* result in a similar chromosome-wide delayed replication timing phenotype. These parallels between ASAR and *Xist* mutation phenotypes suggest a shared mechanism for controlling replication timing in cis.

The formation of liquid-liquid phases via ASAR RNA-RBP interactions is speculative at this point and remains to be tested directly. A full understanding of the mechanisms by which ASARs control chromosome replication timing will require an extensive biochemical and biophysical characterization of the ASAR RNA-protein and protein-protein complexes and how they interact with the DNA replication machinery.

## Materials and methods

**Key resources table**

| Reagent type (species) or resource | Designation | Source or reference | Identifiers | Additional information |
|---|---|---|---|---|
| Antibody | Anti-HLTF (rabbit polyclonal) | Sigma | RRID:AB_1857263 | 1:1000 |
| Antibody | Anti-HNRNPA1 (rabbit polyclonal) | Sigma | HPA004276 | 1:500 |
| Antibody | Anti-HNRNPC (rabbit polyclonal) | Sigma | RRID:AB_2681335 | 1:1000 |
| Antibody | Anti-HNRNPL (rabbit polyclonal) | Sigma | RRID:AB_2681588 | 1:500 |
| Antibody | Anti-HNRNPM (rabbit polyclonal) | Sigma | RRID:AB_1850879 | 1:2000 |
| Antibody | Anti-HNRNPU (rabbit polyclonal) | Sigma | RRID:AB_2683796 | 1:500 |
| Antibody | Anti-HNRNPUL1 (rabbit polyclonal) | Sigma | RRID:AB_2679616 | 1:1000 |
| Antibody | Anti-KHDRBS1 (rabbit polyclonal) | Sigma | RRID:AB_2681417 | 1:500 |
| Antibody | Anti-KHSRP (rabbit polyclonal) | Sigma | RRID:AB_10601582 | 1:500 |
| Antibody | Anti-PTPB1 (mouse monoclonal) | Invitrogen | RRID:AB_2533082 | 1:250 |
| Antibody | Anti-SAFB (rabbit polyclonal) | Sigma | RRID:AB_1850622 | 1:500 |
| Antibody | Anti-SAFB2 (rabbit polyclonal) | Sigma | RRID:AB_2681266 | 1:500 |
| Antibody | Anti-UCHL5 (rabbit polyclonal) | Sigma | RRID:AB_1858585 | 1:2000 |
| Antibody | Anti-alpha-tubulin (mouse monoclonal) | Sigma | RRID:AB_2617116 | 1:10,000 |
| Antibody | Anti-BrdU FITC (mouse monoclonal) | Roche | RRID:AB_11042627 | 50 µg/ml |

*Continued on next page*

*Continued*

| Reagent type (species) or resource | Designation | Source or reference | Identifiers | Additional information |
|---|---|---|---|---|
| Antibody | Anti-mouse IgG, Dylight 680 (goat polyclonal) | Invitrogen | RRID:AB_1965956 | 1:10,000 |
| Antibody | Anti-mouse IgG, Dylight 650 (goat polyclonal) | Thermo Fisher | RRID:AB_10942301 | 1:10,000 |
| Antibody | Anti-rabbit IgG, Dylight 800 (goat polyclonal) | Thermo Fisher | RRID:AB_2556775 | 1:10,000 |
| Antibody | Anti-rabbit IgG, Alexa Fluor Pus 405 (goat polyclonal) | Invitrogen | RRID:AB_2890548 | 1:100 |
| Antibody | Anti-goat IgG, Alexa Fluor 594 (rat monoclonal) | Invitrogen | RRID:AB_2536080 | 1:100 |
| Recombinant DNA reagent | shRNA:*HLTF* (plasmid) | Sigma | TRCN0000272562 | |
| Recombinant DNA reagent | shRNA:*HNRNPA1* (plasmid) | Sigma | TRCN0000006586 | |
| Recombinant DNA reagent | shRNA:*HNRNPC* (plasmid) | Sigma | TRCN0000006644 | |
| Recombinant DNA reagent | shRNA:*HNRNPL* (plasmid) | Sigma | TRCN0000017246 | |
| Recombinant DNA reagent | shRNA:*HNRNPM* (plasmid) | Sigma | TRCN0000314949 | |
| Recombinant DNA reagent | shRNA:*HNRNPU* (plasmid) | Sigma | TRCN0000350256 | |
| Recombinant DNA reagent | shRNA:*HNRNPUL1* (plasmid) | Sigma | TRCN0000074732 | |
| Recombinant DNA reagent | shRNA:*KHDRBS1* (plasmid) | Sigma | TRCN0000428104 | |
| Recombinant DNA reagent | shRNA:*KHDRBS1* (plasmid) | Sigma | TRCN0000428752 | |
| Recombinant DNA reagent | shRNA:*KHSRP* (plasmid) | Sigma | TRCN0000013256 | |
| Recombinant DNA reagent | shRNA:*PTBP1* (plasmid) | Sigma | TRCN0000231420 | |
| Recombinant DNA reagent | shRNA:*SAFB* (plasmid) | Sigma | TRCN000022099 | |
| Recombinant DNA reagent | shRNA:*SAFB2* (plasmid) | Sigma | TRCN0000075057 | |
| Recombinant DNA reagent | shRNA:*UCHL5* (plasmid) | Sigma | TRCN0000234907 | |

## Cell culture

HTD114 cells are a human *APRT* deficient cell line derived from HT1080 cells (*Zhu et al., 1994*), we have extensively genotyped these cells using the Affymetrix Genome-Wide Human SNP Array 6.0 and by whole genome sequencing to generate haplotype phased RNAseq and Repliseq data sets (*Heskett et al., 2020*; *Heskett et al., 2022*; *van Koningsbruggen et al., 2010*). Mouse C2C12 cells were obtained from ATCC and were used for the transgene integration assays, transfected using Lipofectamine 2000 (Invitrogen) according to the manufacturer's recommendations, and clones containing transgene integrations were isolated following selection in media containing G418. HTD114 and C2C12 cells were grown in DMEM (Gibco) supplemented with 10% fetal bovine serum (Hyclone). K562 cells were obtained from ATCC, and cultured in RPMI 1640 (Gibco) supplemented with 10% fetal bovine serum (Hyclone). All cells were confirmed to be mycoplasma free using the LookOut Mycoplasma PCR Detection Kit from Sigma-Aldrich.

## eCLIP analysis

eCLIP data was obtained from *Van Nostrand et al., 2020*. Given that non-coding genes and TLs are expressed at a lower level than coding genes, a sensitive filtering approach was used as recommended by the authors (*Van Nostrand et al., 2020*). We filtered out peaks containing an eCLIP peak p-value greater than p=0.05, and filtered out peaks containing a fold enrichment value less than two.

## Identification of TL

Nuclear-enriched, strand-specific, ribo-minus, total-RNA sequencing libraries were first aligned to the hg19 reference genome using default parameters with STAR (*Dobin et al., 2013*). Next SAMtools (*Li et al., 2009*) was used to remove duplicate and low quality ( ≤ MAPQ20) reads. TLs were defined using a strategy of sequential merging of strand-specific, contiguous intergenic reads, as performed in *Heskett et al., 2022*. Utilizing strand-specific reads, stranded reads separated by 1000 bp or less were merged to create contigs, and the contigs merged again while allowing for gaps of 7 kb to allow for regions that are not uniquely mappable due to presence of full-length LINE elements. TL were then classified above a minimum cutoff length of 50 kb of strand specific, contiguous expression.

## shRNA depletion of RBPs

Using Lipofectamine 2000 (Invitrogen), according to the manufacturer's recommendations, we co-transfected HTD114 cells with plasmids encoding puromycin resistance (PGK-PURO) and shRNAs against 14 different RBPs (see Key Resource Table). Twenty-four hr after transfection, cells were plated in media containing 10 µg/ml puromycin (Millipore/Sigma), and 24 –48 hr following puromycin selection cells were processed for western blot, RNA-DNA FISH or for replication timing assays (see below).

## Western blot analysis

Cells were pelleted, rinsed once in PBS, and lysed by resuspension in RIPA Solubilization Buffer (25 mM Tris-HCl pH 7.6, 150 mM NaCl, 5 mM EDTA,1% NP-40 or 1% Triton X-100, 1% sodium deoxycholate, 0.1% SDS). Protein concentration was determined via the Qubit Protein BR Assay kit (Invitrogen, A50669) on a Qubit 4 Fluorometer. Lysates from HTD114 (20 µg) or K562 (25 µg) cell lines in 1 X Laemmli Sample Buffer (Bio-Rad, 161–0737) with 5% β-mercaptoethanol were loaded onto 10% Mini-PROTEAN TGX Precast Tris-Glycine-SDS polyacrylamide gels, and run in 1 X SDS PAGE running buffer (Bio-Rad, 161–0732) at 100 Volts for 1.25 hr. Proteins were transferred to either NitroBind nitrocellulose (GE, EP4HY00010) or Immobilon-FL PVDF (Millipore, IPFL00010) membranes in Tris-Glycine transfer buffer (25 mM Tris, 192 mM glycine, 20% methanol pH 8.3) at 100 V for 1 hr. All primary antibodies are described in Key Resource Table.

For protein detection, membranes were blocked for one hour in 50% AquaBlock (Arlington Scientific, PP82P-500-QT-1582) in PBS (AquaBlock-P) and probed with specific antibodies (Key Resource Table) in AquaBlock-P with 0.1% Tween 20 (AquaBlock-PT) at 4 °C overnight. The membranes were subjected to four 5 min washes in PBS with 0.1% Tween 20 (PBS-T) at room temperature. Membranes were then incubated with secondary antibodies in AquaBlock-PT for 1 hr at room temperature, and subjected to four washes in PBS-T as described above. Antibody labeling was detected using an Azure Sapphire Biomolecular Imager.

## Immunofluorescence

Cells were allowed to attach to poly-L lysing coated glass slides for 2 hr. Cells were fixed with 4% paraformaldehyde in PBS pH 7.4 for 10 min at room temperature, then washed 3 X with ice-cold PBS. Cells were Permeabilized for 10 min with PBS containing either 0.1% Triton X-100 (PBST), and washed 3 X with PBS at room temperature. Prior to antibody addition, slides were incubated for 30 min with 10% serum from the species the secondary antibody was raised in. Slides were incubated in diluted primary antibody in 1% BSA in PBST in a humidified chamber overnight at 4 °C. Slides were washed three times in PBS, 5 min each wash. Slides were incubated with secondary antibody in 1% BSA for 1 hr at room temperature in the dark. Slides were wash three times with PBS for 5 min each in the dark. All primary antibodies are described in the Key Resource Table.

## CRISPR/Cas9 engineering

Using Lipofectamine 2000 (Invitrogen), according to the manufacturer's recommendations, we co-transfected HTD114 cells with plasmids encoding GFP, sgRNAs and Cas9 endonuclease (Origene). Each plasmid encoded sgRNA was designed to bind at the indicated locations (**Figure 1—figure supplement 1**). 48 h after transfection, cells were plated at clonal density and allowed to expand for 2–3 weeks. The presence of deletions was identified by PCR using the primers described in **Figure 1—figure supplement 1**. The single cell colonies that grew were analyzed for heterozygous deletions by PCR (**Figure 1—figure supplement 1**). We used retention of a heterozygous SNP to identify the disrupted allele, and homozygosity at the SNP confirmed that cell clones were homogenous.

## PCR analysis

Genomic DNA was isolated from tissue culture cells using TRIzol Reagent (Invitrogen). PCR was performed in a 12.5 µL volume using 50–100 ng of genomic DNA, 1 x Standard Taq Buffer (New England Biolabs, Inc), 200 µM each deoxynucleotide triphosphates, 0.2 µM of each primer, and 3 units of Taq DNA Polymerase (New England Biolabs, Inc) under the following reaction conditions: 95 °C for 2 min, followed by 30–40 cycles of 95 °C for 30 s, 55–62°C for 45 s, and 72 °C for 1 min, with a final extension time of 10 min at 72 °C. PCR products were separated on 1% agarose gels, stained with ethidium bromide, and photographed under ultraviolet light illumination. Sequencing of PCR products was carried out at the Vollum Institute DNA Sequencing Core facility. All PCR primers are described in **Figure 1—source data 3**.

## DNA FISH

Mitotic chromosome spreads were prepared as described previously (**Smith et al., 2001**). After RNase (100 µg/ml) treatment for 1 hr at 37 °C, slides were washed in 2XSSC and dehydrated in an ethanol series and allowed to air dry. Chromosomal DNA on the slides was denatured at 75 °C for 3 minutes in 70% formamide/2XSSC, followed by dehydration in an ice-cold ethanol series and allowed to air dry. BAC and Fosmid DNAs were labeled using nick translation (Vysis, Abbott Laboratories) with Spectrum Orange-dUTP, Spectrum Aqua-dUTP or Spectrum Green-dUTP (Vysis). Final probe concentrations varied from 40 to 60 ng/µl. Centromeric probe cocktails (Vysis) and/or whole chromosome paint probes (Metasystems) plus BAC or Fosmid DNAs were denatured at 75 °C for 10 minutes and prehybridized at 37 °C for 10 minutes. Probes were applied to denatured slides and incubated overnight at 37 °C. Post-hybridization washes consisted of one 3 minute wash in 50% formamide/2XSSC at 40 °C followed by one 2 min rinse in PN (0.1 M Na$_2$HPO$_4$, pH 8.0/2.5% Nonidet NP-40) buffer at RT. Coverslips were mounted with Prolong Gold antifade plus DAPI (Invitrogen) and viewed under UV fluorescence (Olympus). All BAC and Fosmid probes are described in **Figure 1—source data 3**.

## RNA-DNA FISH

Cells were plated on Poly-L-Lysine coated (Millipore Singa) glass microscope slides at ~50% confluence and incubated for 4 hr in complete media in a 37 °C humidified CO$_2$ incubator. Slides were rinsed 1 X with sterile RNase-free PBS. Cell Extraction was carried out using ice cold solutions as follows: Slides were incubated for 30 s in CSK buffer (100 mM NaCl/300 mM sucrose/3 mM MgCl$_2$/10 mM PIPES, pH 6.8), 10 min in CSK buffer/0.1% Triton X-100, followed by 30 s in CSK buffer. Cells were then fixed in 4% paraformaldehyde in PBS for 10 min and stored in 70% EtOH at –20 °C until use. Just prior to RNA FISH, slides were dehydrated through an EtOH series and allowed to air dry. Denatured probes were prehybridized at 37 °C for 10 min, applied to non-denatured slides and hybridized at 37 °C for 14–16 hr. Post-hybridization washes consisted of one 3-min wash in 50% formamide/2XSSC at 40 °C followed by one 2-min rinse in 2XSSC/0.1% TX-100 for 1 min at RT. Slides were then fixed in 4% paraformaldehyde in PBS for 5 min at RT, and briefly rinsed in 2XSSC/0.1% TX-100 at RT. Coverslips were mounted with Prolong Gold antifade plus DAPI (Invitrogen) and slides were viewed under UV fluorescence (Olympus). Z-stack images were generated using a Cytovision workstation. After capturing RNA FISH signals, the coverslips were removed, the slides were dehydrated in an ethanol series, and then processed for DNA FISH, beginning with the RNase treatment step, as described above. All BAC and Fosmid probes are described in **Figure 1—source data 3**.

## Chromosome replication timing assay

The BrdU replication timing assay was performed as described previously on exponentially dividing cultures and asynchronously growing cells (*Smith and Thayer, 2012*). Mitotic chromosome spreads were prepared and DNA FISH was performed as described above. The incorporated BrdU was then detected using a FITC-labeled anti-BrdU antibody (Roche). Coverslips were mounted with Prolong Gold antifade plus DAPI (Invitrogen), and viewed under UV fluorescence. All images were captured with an Olympus BX Fluorescent Microscope using a 100 X objective, automatic filter-wheel and Cytovision workstation. Individual chromosomes were identified with either chromosome-specific paints, centromeric probes, BACs or by inverted DAPI staining. Utilizing the Cytovision workstation, each chromosome was isolated from the metaphase spread and a line drawn along the middle of the entire length of the chromosome. The Cytovision software was used to calculate the pixel area and intensity along each chromosome for each fluorochrome occupied by the DAPI and BrdU (FITC) signals. The total amount of fluorescent signal in each chromosome was calculated by multiplying the average pixel intensity by the area occupied by those pixels. The BrdU incorporation into human chromosomes containing CRISPR/Cas9 modifications was calculated by dividing the total incorporation into the chromosome with the deleted chromosome divided by the BrdU incorporation into the non-deleted chromosome within the same cell. Boxplots were generated from data collected from 8 to 12 cells per clone or treatment group. Differences in measurements were tested across categorical groupings by using the Kruskal-Wallis test (*Kruskal, 1964*) and listed as p-values for the corresponding plots. All BAC and Fosmid probes are described in *Figure 1—source data 3*.

## Quantification and statistical analysis

P-values were generated for eCLiP data using the 'region-based' method described in *Van Nostrand et al., 2016*. For 10 kb sliding windows across the genome the log2-Ratio was calculated between the number of reads in the eCLiP sample and matched control, and p-values were generated using the Python scipy.stats function 'zscore'. FDR correction was performed using the Python Statsmodels function 'fdrcorrection'.

## Resource availability

Further information and requests for resources and reagents should be directed to and will be fulfilled by the lead contact, Mathew Thayer (thayerm@ohsu.edu).

## Materials availability statement

All data generated or analyzed during this study are included in the manuscript and supporting files.

## Acknowledgements

MJT was supported by NIH NIGMS (R01GM114162 and R01GM130703). MBH was supported by NIH NCI 4K00CA245677-03.

# Additional information

## Funding

| Funder | Grant reference number | Author |
| --- | --- | --- |
| National Institute of General Medical Sciences | R01GM114162 | Mathew Thayer |
| National Institute of General Medical Sciences | R01GM130703 | Mathew Thayer |

The funders had no role in study design, data collection and interpretation, or the decision to submit the work for publication.

## Author contributions
Mathew Thayer, Conceptualization, Resources, Data curation, Formal analysis, Supervision, Funding acquisition, Validation, Investigation, Visualization, Methodology, Writing – original draft, Project administration, Writing – review and editing; Michael B Heskett, Conceptualization, Data curation, Software, Formal analysis, Validation, Investigation, Visualization, Methodology, Writing – original draft, Writing – review and editing; Leslie G Smith, Data curation, Visualization; Paul T Spellman, Conceptualization, Data curation, Software, Funding acquisition; Phillip A Yates, Conceptualization, Validation, Investigation, Visualization, Methodology, Writing – original draft, Writing – review and editing

## Author ORCIDs
Mathew Thayer ⓘ https://orcid.org/0000-0001-6483-1661
Michael B Heskett ⓘ http://orcid.org/0000-0003-2089-3910
Phillip A Yates ⓘ http://orcid.org/0000-0003-2016-9789

Reviewer #1 (Public Review): https://doi.org/10.7554/eLife.95898.3.sa1
Reviewer #2 (Public Review): https://doi.org/10.7554/eLife.95898.3.sa2
Author response https://doi.org/10.7554/eLife.95898.3.sa3

---

# Additional files

## Supplementary files
• MDAR checklist

## Data availability
All data generated or analyzed during this study are included in the manuscript and supporting files; source data files have been provided for Figures 1, 2 and 3.

The following previously published datasets were used:

| Author(s) | Year | Dataset title | Dataset URL | Database and Identifier |
|---|---|---|---|---|
| Yeo G | 2020 | Summary for publication file set ENCSR456FVU | https://www.encodeproject.org/publication-data/ENCSR456FVU/ | ENCODE, ENCSR456FVU |
| Yeo G | 2020 | Best practices for eCLIP experiments and analysis | https://www.ncbi.nlm.nih.gov/geo/query/acc.cgi?acc=GSE107768 | NCBI Gene Expression Omnibus, GSE107768 |

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
