## [Editor Report · eLife assessment]

This **important** study expands generally upon our understanding of the role of hnRNP proteins in lncRNA function through analysis of ASAR genes that are present on all chromosomes and of profound significance. The findings provide **convincing** evidence linking ASARs with the phenomenon of RNA retention on chromosomes, including X inactivation, thereby providing an expanded context for studies in these areas. This manuscript will be of interest to researchers studying gene regulation and the interactions and functional roles of hnRNP and lncRNAs.

---

## [Referee Report · Reviewer #1 (Public Review)]

Summary:

Thayer et al build upon their prior findings that ASAR long noncoding RNAs (lncRNAs) are chromatin-associated and are implicated in control of replication timing. To explore the mechanism of function of ASAR transcripts, they leveraged the ENCODE RNA binding protein eCLIP datasets to show that a 7kb region of ASAR6-141 is bound by multiple hnRNP proteins. Deletion of this 7kb region resulted in delayed chromosome 6 replication. Furthermore, ectopic integration of the ASAR6-141 7kb region into autosomes or the inactive X-chromosome also resulted in delayed chromosome replication. They then use RNA FISH experiments to show that knockdown of these hnRNP proteins disrupts ASAR6-141 localization to chromatin and in turn replication timing.

Strengths:

Given prior publications showing HNRNPU to be important for chromatin retention of XIST and Firre, this work expands upon our understanding on the role of hnRNP proteins in lncRNA function.

Weaknesses:

The work presented is mechanistically interesting, however, one must be careful with the over interpretation that hnRNP proteins can regular chromosome replication directly.

---

## [Referee Report · Reviewer #2 (Public Review)]

Summary:

This paper reports a role for a substantial number of RNA binding proteins (RBPs), in particular hnRNPs, in the function of ASAR "genes". ASARs are (very) long, non-coding RNAs (lncRNAs) that control allelic expression imbalance (e.g.: mono-allelic expression) and replication timing of their resident chromosomes. These relatively novel "genes" have recently been identified on all human autosomes and are of broad significance given their critical importance for basic chromosomal functions and stability. However, the mechanism(s) of ASAR function remain unclear. ASARs exhibit some functional relatedness to Xist RNA, including persistent association of the expressed RNA with its resident chromosome, and similarities in the composition of RNA sequences associated with ASARs, in particular Line1 RNAs. Recent findings that certain hnRNPs control the chromosome territory retention of Cot1-bearing RNAs (which includes Line1) led the authors to test hypothesis that hnRNPs might regulate ASARs.

Specific new findings in this paper:

-Analysis of eCLIP (RNA-protein interaction) ENCODE data shows numerous interactions of the ASAR6-141 RNA with RBPs, including hnRNPs (e.g.: HNRNPU) that have been implicated in the retention of RNAs within local chromosome territories.

-most of these interactions can be mapped to a 7kb region of the 185kb ASAR6-141 RNA

-deletion of this 7kb region is sufficient to induce the DMC/DRT phenotype associated with deletion of the entire ASAR region

-ectopic integration into mouse autosomes of the 7kb region is sufficient to cause DMC/DRT of the targeted autosome, and a similar effect upon ectopic integration into inactive X. This raises the question about integration into the active X, which was not mentioned. Is integration into the active X observed? Is it possible that integration might alter Xist expression confounding this interpretation?

-Knockdown of RBPs that bind the 7kb region causes dissociation of ASAR6-141 RNA from its chromosome territory, and, remarkably, dissociation of Xist RNA from inactive X, and mis-colocalization of the ASAR6-141 and Xist RNAs. Depletion of these RBPs causes DMC/DRT on all autosomes.

Strengths:

These are compelling results suggesting shared mechanism(s) in the regulation of ASARs and Xist RNAs by RBPs that bind Cot1 sequences in these lncRNAs. The identification of these RBPs as shared effectors of ASARs and Xist that are required for RNA territory localization mechanistically links previously independent phenomena.

The data are convincing and support the conclusions. The replication timing method is low resolution and is only a relative measure but seems adequate for the task at hand. The FISH experiments are convincing. The quality of the images is impressive.

Links to other subfields like X-inactivation and RNA association with chromosome territories provide novel context and protein players, new phenotypes to examine

Weaknesses:

The exact effects of knockdown experiments are unclear and may be indirect, which is acknowledged.

The mechanism is not much clearer than before.

---

## [Author Response]

The following is the authors’ response to the original reviews.

**Recommendations for the authors:**

**Reviewer #1 (Recommendations For The Authors):**
Major recommendations(1) In lines 42-44 (abstract), the authors state that "ASARs function as essential RNA scaffolds for the assembly of hnRNP complexes that help maintain the structural integrity of each mammalian chromosome". Similar conclusions are restated in lines 138-140. Based on the data presented, it is evident that ASARs localization on chromatin is dependent on hnRNPs. However, there is insufficient evidence to conclude that ASARs cause the assembly of hnRNP complexes or that these hnRNP complexes are directly responsible for the regulation of chromosome replication. Please revise your claims.

We have modified the text as follows: “Our results further demonstrate the role that ASARs play during the temporal order of genome-wide replication, and we propose that ASARs function as essential RNA scaffolds for the assembly of hnRNP complexes that help maintain the structural integrity of each mammalian chromosome.”

(2) In the analysis in Figure 1C- F, it is unclear why XIST is used as a comparison to ASAR6-141. A more meaningful control would be to show that hnRNPs preferentially bind ASAR6-141 relative to all expressed transcripts. Also, some panels are missing the y-axis label.

We have genetically validated 8 different ASAR genes for their role in controlling chromosome-wide replication timing. The only other gene known to control chromosome-wide replication timing is XIST, which also encodes a chromosome-associated lncRNA. Our analysis of publicly available eCLIP data (and previous literature on XIST-binding proteins) showed substantial overlap between RBPs that associate with ASARs and XIST. Hence, we anticipated that at least some RBP knockdowns would affect both lncRNAs, despite their contrasting functions. In addition, we routinely use XIST RNA as a positive control in RNA FISH assays, as the XIST RNA FISH protocol represents a robust and well validated chromosomal RNA FISH procedure.

y-axis labels have been added to Figure 1.

(3) In Figure 2K&L, it would be beneficial to quantify and normalize the BrdU incorporation, as ectopic integration of the sense 7kb region appears to result in overall higher BrdU incorporation in all chromosomes, not just chromosome 5.

There are two main aspects of the BrdU incorporation assay that we use: (1) The BrdU incorporation banding pattern on each chromosome is unique to that chromosome, and the banding pattern is also representative of the time during S phase when the BrdU incorporation occurred, i.e. we detect a different banding pattern if BrdU is incorporated in early S phase versus late S phase. (2) The amount of BrdU incorporation can be used to measure the synchrony between chromosome homologs, but only within the same cell. Thus, we generate a ratio of BrdU incorporation in chromosome homologs in individual cells, then compare the ratio of incorporation into each chromosome pair in multiple cells (see Figure 2B-E). The overall BrdU incorporation into the chromosomes of different cells is quite variable; however, the banding pattern and ratio of BrdU incorporation in chromosome homologs in individual cells is comparable, unless we have disrupted or ectopically integrated an ASAR. Given the variability in overall BrdU incorporation detected between different cells in the population this is not a useful readout for measuring synchronous versus asynchronous replication between chromosome homologs.

(4) hnRNP protein can regulate multiple aspects of RNA processing other than chromatin retention. Hence, it would be beneficial to rule out an alternative hypothesis as to what the hnRNP knockdowns do to ASAR6-131? For example, assessing changes in RNA levels or splicing upon knockdown of hnRNPs using qPCR?

We agree that direct roles for any of the hnRNP/RBPs that are critical for ASAR RNA localization and replication timing have not been established. However, our findings combined with the observation that cells depleted of HNRNPU show reduced origin licensing in G1, and show reduced origin activation frequency during S phase (PMID: 34888666), supports a role for HNRNPU, either directly or indirectly, in DNA replication. Furthermore, we also found that depletion of the DNA replication fork remodeler HLTF or the deubiquitinase UCHL5 also results in mis-localization of ASAR RNAs, and results in asynchronous replication of every autosome pair, indicating that ASAR RNA mis-localization and asynchronous replication are not simply a phenotype associated with hnRNP depletions. A full mechanistic understanding of the role that ASAR RNAs play in combination with this relatively large and diverse set of hnRNP/RBPs will require a better understanding of the direct roles that each protein, and any higher order complexes that contain these proteins, play in regulating DNA synthesis, splicing, transcription, chromatin structure and/or ASAR RNA localization.

(5) Both the disruption and ectopic expression of the 7kb region result in delayed chromosome replication. Would one not expect there to be opposing effects on replication timing? Please discuss.

One puzzling set of observations is that loss of function mutations and gain of function mutations of ASAR genes result in a similar delayed replication timing and delayed mitotic condensation phenotype. We have detected delayed replication timing in human cells following genetic knockouts (loss of function) of eight different ASAR genes located on 5 different autosomes. We have also detected delayed replication timing on mouse chromosomes expressing transgenes (gain of function) from three different ASAR genes (ASAR6, ASAR6-141, and ASAR15). The ASAR transgenes ranged in size from an ~180kb BAC, to an ~3kb PCR product. One possible explanation for these observations is that ectopic integration of ASAR transgenes function in a dominant negative manner by interfering with the endogenous “ASARs” on the integrated chromosomes. Consistent with this possibility is that we recently identified ASAR candidate genes on every human autosome (PMC9588035). Our favored model is that expression of ASAR transgenes integrated into mouse chromosomes disrupts the function of endogenous ASARs by "out-competing" them for shared RBPs. We also point out that a similar ectopic integration assay, using *Xist* transgenes, has been an informative assay for characterization of *Xist* functions, including the ability to delay replication timing and induce gene silencing on autosomes (reviewed in PMID:19898525). One intriguing observation (yet largely ignored by the X inactivation field) is that deletion of the Xist gene on either the active or inactive X chromosomes in somatic cells results in delayed replication timing of the X chromosomes (PMC1667074; PMC1456779). Thus, both loss of function and gain of function mutations of Xist result in a similar delayed replication timing phenotype. Given these parallels between Xist and ASAR gene mutation phenotypes we were curious to test the consequences of ASAR gain of function on the inactive X chromosome. In this manuscript, we integrated the ~7kb ASAR6-141 transgene into the inactive X chromosome, and detected a delayed replication timing phenotype on the integrated X chromosome. We also detected an association between Xist and ASAR RNAs using RNA FISH in interphase cells (Figure 4A and 4B), which supports the observations that ASAR RNAs and XIST RNA are bound by a partially overlapping set of hnRNP/RBPs (Figure 1D-F), and is consistent with the model that ASAR transgenes disrupt function by competition for shared RBPs. Dissecting the roles that the hnRNP/RBPs that interact with both ASAR and XIST RNAs will undoubtably give important insights into both XIST and ASAR function, and how these poorly understood chromosomal phenotypes are generated.

Minor recommendations(1) In Figure 1G, it would be informative to show where the LINE-1 element within ASAR6-141 is located to get a sense of what hnRNP proteins bind to it.

There are numerous LINE-1 elements within the ASAR6-141 gene. The ~7kb RBPD does not contain LINE-1 sequences. Therefore, we did not detect significant hnRNP/RBP eCLIP peaks within LINE-1 sequences.

(2) The rationale for ectopic integration of the 7kb region into the inactive X-chromosome is unclear. Is there something unique about the replication of the inactive X or were you interested in seeing whether the 7kb region could escape X-inactivation?

Given the parallels between Xist and ASAR gene mutation phenotypes, i.e. loss of function and gain of function result in delayed replication timing (see above), we were curious to test the consequences of ASAR gene gain of function on the inactive X chromosome. One possibility was reversal of X inactivation and a shift to earlier replication timing. However, we detected delayed replication timing on the inactive X, and an enhanced XIST RNA FISH signal that overlapped with the ASAR RNA. This speaks to the comment of Reviewer 2 questioning: "Is it possible that integration might alter Xist expression confounding this interpretation? ". The enhanced XIST RNA FISH signal suggests that the delayed replication of the inactive X is not due to reduced expression of XIST RNA.